# MAP7 regulates axon morphogenesis by recruiting kinesin-1 to microtubules and modulating organelle transport

Stephen R Tymanskyj[1,2], Benjamin H Yang[1,2], Kristen J Verhey[3], Le Ma[1,2]*

[1]Jefferson Synaptic Biology Center, Vickie and Jack Farber Institute for Neuroscience, Thomas Jefferson University, Philadelphia, United States; [2]Department of Neuroscience, Sidney Kimmel Medical College, Thomas Jefferson University, Philadelphia, United States; [3]Department of Cell and Developmental Biology, University of Michigan Medical School, Ann Arbor, United States

**Abstract** Neuronal cell morphogenesis depends on proper regulation of microtubule-based transport, but the underlying mechanisms are not well understood. Here, we report our study of MAP7, a unique microtubule-associated protein that interacts with both microtubules and the motor protein kinesin-1. Structure-function analysis in rat embryonic sensory neurons shows that the kinesin-1 interacting domain in MAP7 is required for axon and branch growth but not for branch formation. Also, two unique microtubule binding sites are found in MAP7 that have distinct dissociation kinetics and are both required for branch formation. Furthermore, MAP7 recruits kinesin-1 dynamically to microtubules, leading to alterations in organelle transport behaviors, particularly pause/speed switching. As MAP7 is localized to branch sites, our results suggest a novel mechanism mediated by the dual interactions of MAP7 with microtubules and kinesin-1 in the precise control of microtubule-based transport during axon morphogenesis.
DOI: https://doi.org/10.7554/eLife.36374.001

*For correspondence:
le.ma@jefferson.edu

**Competing interests:** The authors declare that no competing interests exist.

## Introduction

Neuronal cell functions, including axon morphogenesis, require proper regulation of intracellular transport on microtubules via a variety of motor and non-motor microtubule-associated proteins (MAPs) (*Armijo-Weingart and Gallo, 2017*; *Kalil and Dent, 2014*; *Kapitein and Hoogenraad, 2015*; *Kevenaar and Hoogenraad, 2015*). Motor proteins consume ATP to move membrane or protein cargos along microtubules, whereas non-motor MAPs often modulate microtubule assembly dynamics and stability (*Kapitein and Hoogenraad, 2015*). Neurons are enriched with lattice-binding non-motor MAPs that are often associated with stable microtubules in axons and dendrites (*Kapitein and Hoogenraad, 2015*). Some of these proteins can influence motor protein functions (*Encalada and Goldstein, 2014*; *Fu and Holzbaur, 2014*; *Nirschl et al., 2017*). For example, the protein tau can inhibit motor function of the conventional kinesin (kinesin-1) by dissociating it from microtubules *in vitro* (*Dixit et al., 2008*). However, the mechanism and the functional role of the interaction between motor and non-motor MAPs in neurons remain poorly understood.

We address this question by studying MAP7 (also known as ensconsin or EMAP-115), a non-motor MAP, for its unique interaction with both microtubules and the kinesin-1 motor. MAP7 was identified from HeLa cell lysates based on its ability to bind microtubules (*Bulinski and Bossler, 1994*; *Masson and Kreis, 1993*). It is expressed in many cell types and involved in many cellular processes. In *Drosophila*, MAP7 has been shown to regulate cell polarity of oocytes, nuclear positioning in muscle cells, organelle transport in S2 cells, and spindle morphogenesis in neural stem cells (*Barlan et al., 2013*; *Gallaud et al., 2014*; *Metzger et al., 2012*; *Sung et al., 2008*). In mice,

mutations in MAP7 are linked to spermatogenesis and histocompatibility (*Komada et al., 2000*; *Magnan et al., 2009*). Recently, we described the first neuronal function of MAP7 in regulating axon collateral branch development of rodent sensory neurons in the dorsal root ganglion (DRG) (*Tymanskyj et al., 2017*). However, the mechanisms mediating MAP7 function in neurons are not known.

MAP7 consists of three domains (*Figure 1A*): an amino (N) domain containing the first coiled-coil (CC1) that binds microtubules (*Masson and Kreis, 1993*), a middle P domain that contains many potential phosphorylation sites (*Masson and Kreis, 1995*), and a carboxyl domain (C) domain containing the second coiled-coil (CC2) that interacts with kinesin-1 (*Metzger et al., 2012*). Studies in *Drosophila* cells have shown that deletion of the C domain affects kinesin-based cell polarity, nuclear migration, organelle transport, and spindle segregation (*Barlan et al., 2013*; *Gallaud et al., 2014*; *Metzger et al., 2012*; *Sung et al., 2008*), suggesting a functional role of the MAP7-kinesin interaction. *In vitro* data have suggested that MAP7 recruits kinesin-1 to microtubules (*Monroy et al., 2018*; *Sung et al., 2008*), but the exact impact of this recruitment on kinesin-1-mediated transport is not completely understood. Nevertheless, the ability of MAP7 to recruit kinesin-1 to microtubules suggests an intriguing function in regulating kinesin-mediated transport in neurons, especially during axon morphogenesis.

Axon morphogenesis involves axon growth and branching (*Gibson and Ma, 2011*; *Kalil and Dent, 2014*). Axon branching can be further divided into branch formation and branch growth. These processes involve the regulation of microtubule assembly/stability by non-motor MAPs and organelle transport by motor proteins (*Armijo-Weingart and Gallo, 2017*; *Baas et al., 2016*; *Dent et al., 2011*; *Gallo, 2011*). For example, kinesin-1 stabilizes axonal arbors and promotes branch growth (*Seno et al., 2016*); and non-motor MAPs, including doublecortin and MAP1B, appear to suppress branch formation (*Barnat et al., 2016*; *Bilimoria et al., 2010*; *Bouquet et al., 2004*; *Tymanskyj et al., 2012*). However, how motor and non-motor MAPs coordinate to regulate these processes is largely unknown (*Armijo-Weingart and Gallo, 2017*; *Gallo, 2011*; *Gallo and Letourneau, 1999*; *Lewis et al., 2013*). The newly identified role of MAP7 in branch development of DRG axons (*Tymanskyj et al., 2017*), along with its ability to interact with both microtubules and kinesin-1, suggests a potential mechanism.

To understand the cooperation of MAP7 and kinesin-1 during axon morphogenesis, we first determined the MAP7 domains that are required for axon growth and branching using a gain-of-function approach in a well-defined rat DRG neuronal cell culture (*Tymanskyj et al., 2017*; *Wang et al., 1999*; *Zhao et al., 2009*). We then analyzed these domains in COS cells and characterized their microtubule binding properties as well as their roles in recruiting kinesin-1 to microtubules. Finally, using an optogenetic tool that directly loads active kinesin-1 motors to a specific organelle surface (*van Bergeijk et al., 2015*), we show that MAP7 modulates kinesin-1-mediated organelle transport by increasing pause/speed switching in COS cells and in DRG axons. Our findings have thus revealed a coordinated interaction between MAP7 and kinesin-1 in regulating axon growth and branching as well as a role of the kinesin-1 recruitment in regulating organelle transport during neuronal cell morphogenesis.

## Results

### MAP7 domains have distinct roles in axon morphogenesis

To understand the contribution of MAP7 domains in axon morphogenesis, we used E14 rat DRG neurons that lack endogenous MAP7 expression at this time point and which exhibit a simple morphology when grown in culture (*Tymanskyj et al., 2017*; *Wang et al., 1999*; *Zhao et al., 2009*) (*Figure 1B*, and *Figure 1—figure supplement 1*). Expression of an EGFP fusion protein of full length (FL) MAP7 (MAP7-FL-EGFP) resulted in a threefold increase in the number of branches produced per cell and a fivefold increase in the number of interstitial branches, defined as those arising from 90% of the axon length proximal to the soma, when compared with the EGFP control (*Figure 1B,C*, *Figure 1—figure supplement 1*). Interestingly, however, comparison of the axon length also revealed that MAP7-FL-EGFP-expressing axons were only half the length of EGFP control axons (*Figure 1D*). These results suggest that MAP7 has opposite effects on axon growth and branch formation.

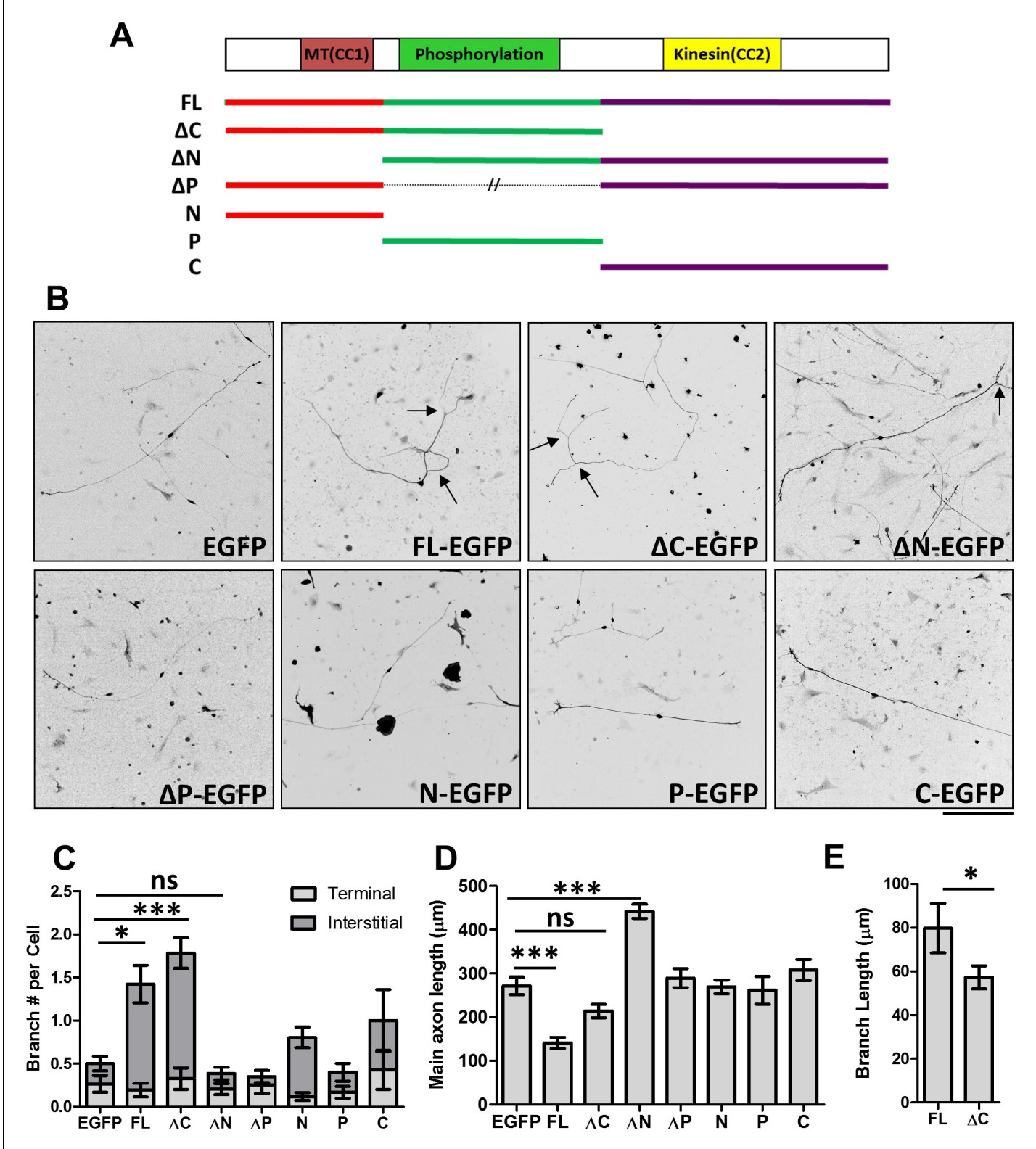

**Figure 1.** Distinct roles of MAP7 domains in DRG axon growth and branching. (**A**) Primary structure of MAP7, indicating the phosphorylation (P) domain and the two coiled-coil (CC) regions that interact with microtubules (MT(CC1)) and kinesin-1 (Kinesin(CC2)). The full length (FL) MAP7 and various fragments used in the study are illustrated by line drawings. (**B**) Representative images of neurofilament staining in E14 rat DRG neurons expressing EGFP or EGFP-tagged fusion proteins of MAP7-FL or various MAP7 fragments. Arrows point to interstitial branches. (**C**) Quantification of the number of branches per cell as measured by counting the total number of tips per neuron in E14 DRG neurons expressing EGFP or EGFP fusion proteins.

*Figure 1 continued on next page*

*Figure 1 continued*

Branches were further divided into two groups: terminal branches arising from the distal 10% part of the axon and interstitial branches arising from the rest of the axons. n = 33, 26, 46, 39, 20, 51, 31, 14 for EGFP, FL, ΔC, ΔN, ΔP, N, P and C respectively. ANOVA-test (Mean ±SEM): EGFP-FL, p=0.013; EGFP-ΔC, p≤0.0001; EGFP-ΔN, p=0.98. (D) Quantification of the total length of main axons in neurons expressing different MAP7 constructs. n = 44, 21, 18, 22, 21, 77, 12, 15 for EGFP, FL, ΔC, ΔN, ΔP, N, P and C respectively. ANOVA-test (Mean ±SEM): EGFP-FL, p=0.0003; EGFP-ΔC, p=0.29; EGFP-ΔN, p≤0.0001. (E) Comparison of the branch length between MAP7-FL-EGFP and MAP7-ΔC-EGFP expressing DRG neurons. n = 36 for FL and 73 for ΔC. T-test (Mean ±SEM): p=0.04. *p<0.05; **p<0.01; ***p<0.001; ns: not significant. Scale bar: 200 μm.

DOI: https://doi.org/10.7554/eLife.36374.002

The following source data and figure supplement are available for figure 1:

**Source data 1.** Data for the measurement of branch number, axon length, and branch length in *Figure 1C–E.*
DOI: https://doi.org/10.7554/eLife.36374.004

**Figure supplement 1.** Traces of axonal morphology shown in *Figure 1B*. Scale bar: 200 μm.
DOI: https://doi.org/10.7554/eLife.36374.003

Using this gain-of-function approach, we tested the requirement of the C-terminal kinesin interacting domain in axon growth and branching. Expression of a ΔC fragment lacking the C domain induced a similar number of branches to the MAP7-FL protein (*Figure 1B,C*, *Figure 1—figure supplement 1*), suggesting that the kinesin interaction is not required for branch formation per se. However, the ΔC fragment did not significantly alter the length of the main axon when compared with the EGFP control (*Figure 1D*). Further analysis revealed that ΔC-expressing neurons also generated shorter branches than those expressing the MAP7-FL protein (*Figure 1E*), supporting the requirement of the kinesin interacting domain for branch growth.

We next tested the requirement for the N domain that has been shown to bind microtubules (*Masson and Kreis, 1993*). Unlike the MAP7-FL protein, expression of an EGFP fusion of the N-deletion fragment (ΔN) did not increase the number of branches (*Figure 1B,C*, *Figure 1—figure supplement 1*), suggesting that the N domain is required for branch formation. Interestingly, however, ΔN-expressing neurons have significantly longer axons compared with neurons expressing the EGFP control or other MAP7 constructs (*Figure 1B,D*), indicating that both P and C domains combined are sufficient to promote axon growth. To test the requirement of the P domain, we generated an EGFP fusion of MAP7 with the P domain deleted. Expression of this construct (ΔP-EGFP) did not increase the number of branches per cell, nor did it increase the length of main axons (*Figure 1B–D*, *Figure 1—figure supplement 1*). This result suggests that the P domain is required for both branch formation and axon growth. Finally, we tested each domain individually and found that none of the domains alone (N, P, or C) were sufficient to promote branch formation or main axon growth (*Figure 1C,D*, *Figure 1—figure supplement 1*). Taken together, these results suggest that MAP7 domains play distinct roles in three intimately related processes during axon morphogenesis: axonal growth requires both P and C domains, branch formation requires both N and P domains, and branch growth requires all three domains.

We also examined the requirement of different domains for MAP7 localization, as antibody staining showed that the endogenous MAP7 was localized to branch sites in E17 rat DRG neurons (*Figure 2A*) when the expression reached its peak (*Tymanskyj et al., 2017*). We quantified the number of branches containing MAP7 and found that 65% of branches have endogenous MAP7 concentrated at the base (*Figure 2B*). Consistent with this, MAP7-FL-EGFP expressed in E14 DRG neurons was also localized to 57% of branch sites but excluded from the nerve terminals, in contrast to the uniform localization of the EGFP control (*Figure 2B,C*). This can be seen by fluorescence line scan along the length of the axon, which reveals decreasing tubulin signals from the cell body to the axon terminal but a peak of MAP7 signals coinciding with a branch in the middle of an axon (asterisk; *Figure 2D*). Like MAP7-FL, the ΔC fragment also accumulated at branch points along the axon and avoided nerve terminals (*Figure 2C*), suggesting that the kinesin interaction is not required for this localization. In contrast, deletion of either N or P domain (ΔN and ΔP) led to the even distribution of EGFP fusion proteins throughout the axon and expanded localization to the axon terminals (*Figure 2C*). Line scans showed an even distribution of the ΔN-EGFP signal, in contrast to the graded tubulin signal that tapered off at the distal end (*Figure 2D*). Furthermore, each single MAP7 domain alone (N, P, and C) behaved like the EGFP control with uniform localization (*Figure 2C* for

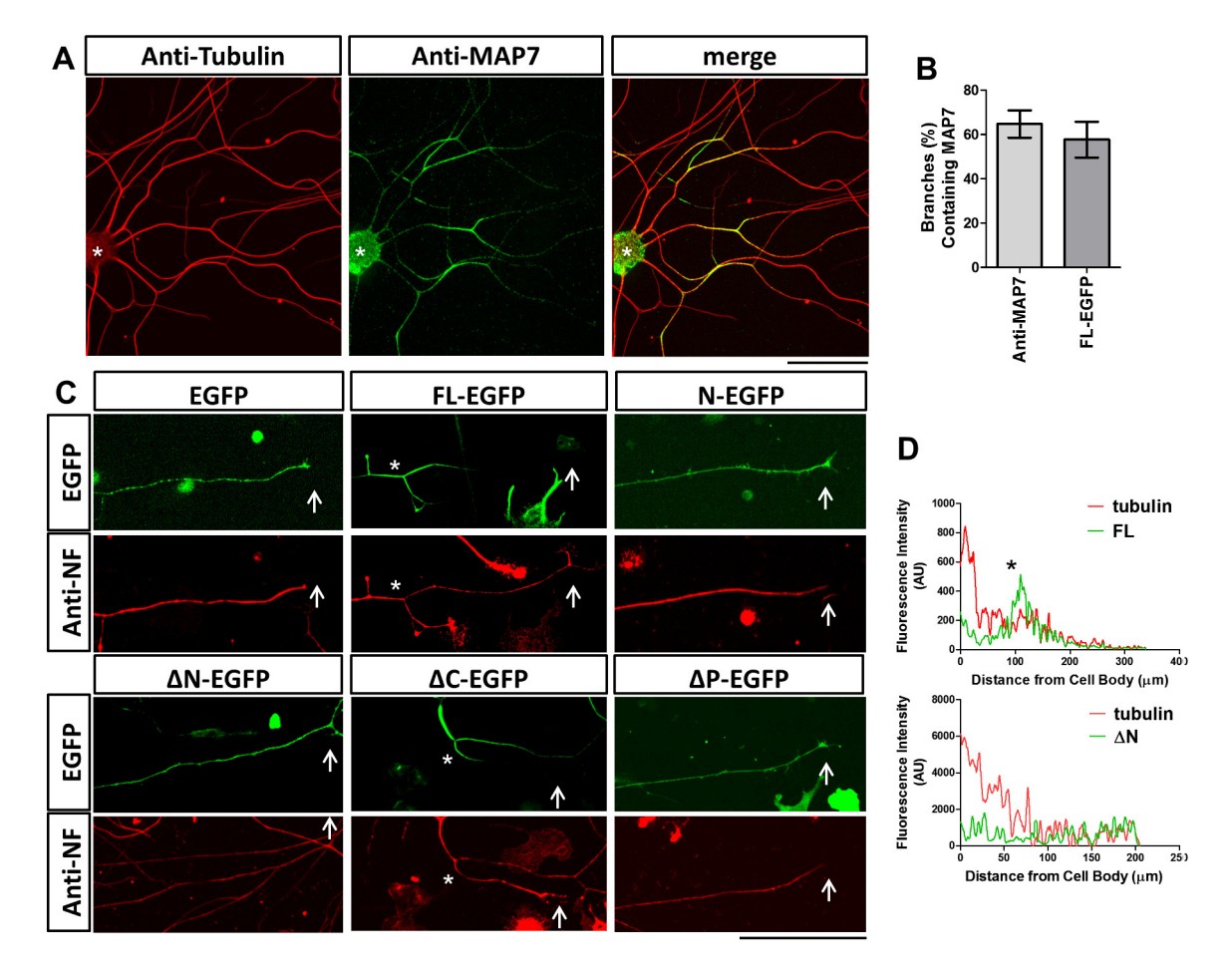

**Figure 2.** Localization of MAP7 in DRG axons. (A) Fluorescence confocal images of E17 rat DRG axons immunostained for tubulin (red) and endogenous MAP7 (green) extending out from an explant (asterisks). The color-merged image is also shown. (B) Quantification of branches with either endogenous MAP7 by antibody staining or overexpressed FL-EGFP concentrated at branch sites. (C) Representative fluorescence images of E14 rat DRG neurons expressing EGFP or EGFP fusion proteins of MAP7-FL or various MAP7 fragments. Green: antibody staining for EGFP; red: antibody staining for neurofilament (NF). Arrows point to axonal terminals and asterisks indicate branch sites. (D) Line scans in axons (cell body, left, growth cone, right) of fluorescence signals from antibody staining for tubulin (red lines) and EGFP fusions (green lines) of FL or ΔN of MAP7. The asterisk indicates the location of a branch.

DOI: https://doi.org/10.7554/eLife.36374.005

The following source data is available for figure 2:

**Source data 1.** Quantification of the percentage of branches containing MAP7 in *Figure 2B*.
DOI: https://doi.org/10.7554/eLife.36374.006

N, and data not shown for P and C). These results demonstrate that the unique localization of MAP7 at branch sites requires both the N and the P domains but not the C domain.

## Identification and characterization of a second microtubule binding site in the P domain of MAP7

Because of the unique functions of each domain in axon morphogenesis, we next determined the domain requirement for the MAP7 interaction with microtubules in live COS cells. Expression of MAP7-FL-EGFP showed localization of the protein to microtubules labeled with tubulin-mCherry (*Figure 3A*). As earlier studies in HeLa cells have suggested that MAP7 binds to microtubules via CC1 in the N domain (*Masson and Kreis, 1993*) (*Figure 1A*), we first tested the EGFP-labeled N domain and found that it indeed bound to microtubules on its own (*Figure 3A*). However,

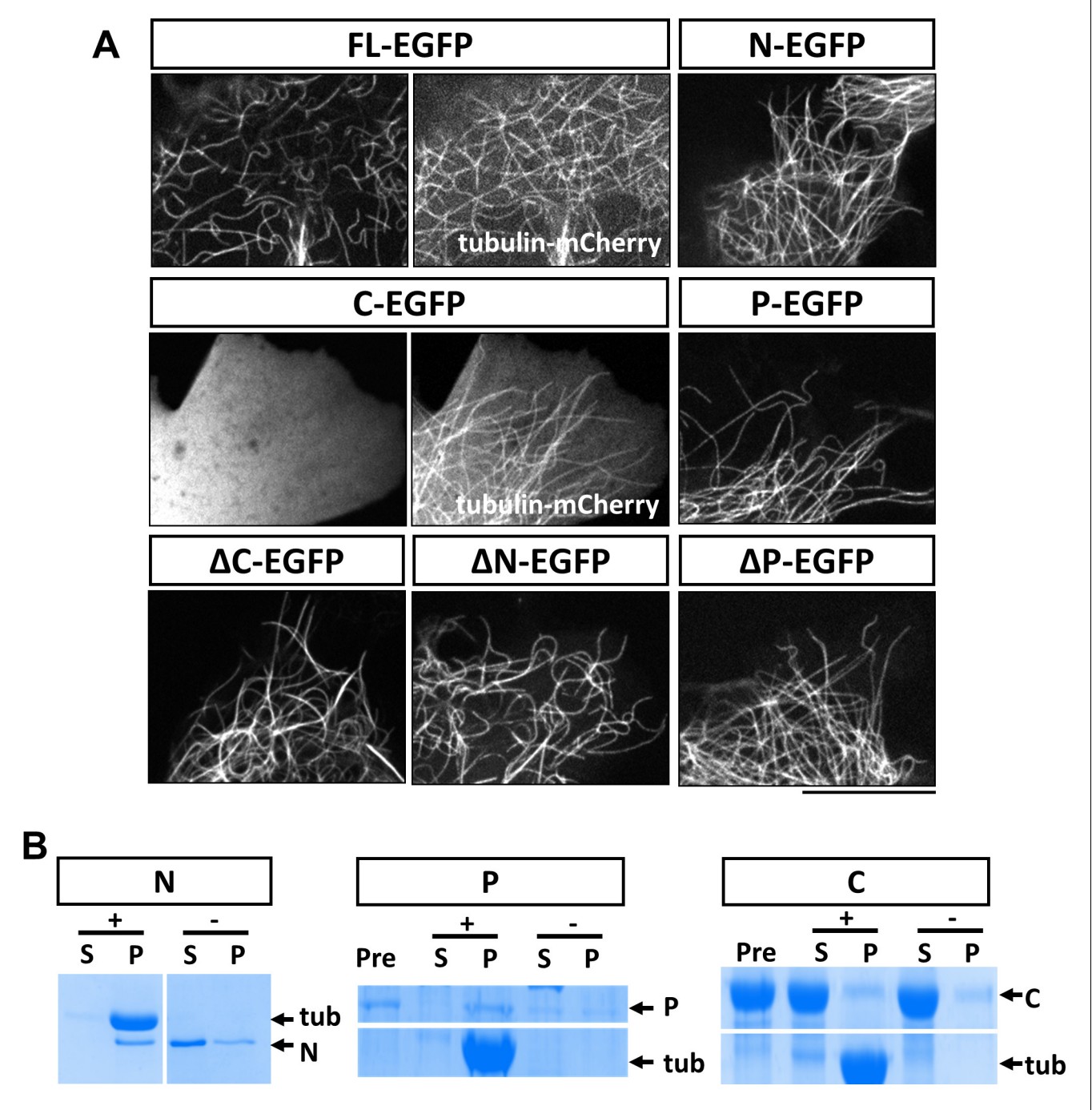

**Figure 3.** Characterization of MAP7 binding to microtubules. (**A**) Confocal fluorescence images in live COS cells expressing EGFP fusion proteins of MAP7-FL or various fragments. Tubulin-mCherry were co-expressed to show microtubules (only for FL-EGFP and C-EGFP). (**B**) Co-sedimentation analysis of different MAP7 domains with (+) or without (-) taxol-stabilized microtubules. Representative Coomassie stained SDS-PAGE gels of the supernatants (S) and pellets (P) along with pre-binding samples (Pre) for purified GST-MAP7-N (N), MAP7-P-mCherry (P) or MAP7-C-mCherry (C). n = 3. Note the presence of tubulin bands only in the pellet of the samples incubated with microtubules.

DOI: https://doi.org/10.7554/eLife.36374.007

The following figure supplements are available for figure 3:

**Figure supplement 1.** Western blot analysis of EGFP fusion proteins of MAP7-FL and various domain fragments expressed in COS cells.
DOI: https://doi.org/10.7554/eLife.36374.008

**Figure supplement 2.** Microtubule binding by individual MAP7 domains in a co-sedimentation assay.
DOI: https://doi.org/10.7554/eLife.36374.009

deletion of the N domain from MAP7 did not eliminate microtubule binding, as the ΔN fragment, when expressed at similar protein levels in COS cells (*Figure 3—figure supplement 1*), also bound microtubules in a manner similar to the MAP7-FL protein (*Figure 3A*). Single domain analysis indicates that the P but not the C domain can also bind microtubules (*Figure 3A*). As a result, the ΔC fragment remained bound to microtubules (*Figure 3A*). Thus, in addition to CC1 in the N domain, a second microtubule binding site is present in the P domain that allows MAP7 to interact with microtubules.

To obtain further support for this idea, we tested whether the P domain binds microtubules directly using a co-sedimentation assay. As a positive control, we generated recombinant GST fusion proteins of the N domain in *E. coli*. After incubation with taxol-stabilized microtubules (20 μM), almost all the purified N protein (2 μM) appeared in the pellet; in contrast, in the absence of microtubules, the N fusion protein largely remained in the supernatant (*Figure 3B*; *Figure 3—figure supplement 2A*). As the P domain fusion protein was not soluble in bacteria, we generated a mCherry fusion protein of the P domain expressed in 293 F cells. As shown in *Figure 3B*, incubation of P-mCherry (2 μM) with microtubules resulted in nearly all P-mCherry proteins co-pelleting with microtubules, whereas in the absence of microtubules, less than half of the total protein was found in the pellet, likely because of protein aggregation (*Figure 3—figure supplement 2B*). As a negative control, an mCherry fusion protein of the C domain (~20 μM) did not co-pellet with microtubules (*Figure 3B*, *Figure 3—figure supplement 2C*). Finally, none of the contaminant proteins from the purification pelleted with microtubules (*Figure 3—figure supplement 2B*), suggesting that the interaction between the P domain and microtubules is specific and direct.

To further understand the nature of this new microtubule binding site, we analyzed the dynamics of microtubule binding by MAP7 domains using fluorescence recovery after photobleaching (FRAP) (*Figure 4A*). COS cells were co-transfected with EGFP fusion proteins of MAP7-FL or MAP7 fragments along with tubulin-mCherry to identify single intact microtubules (data not shown) for FRAP analysis. A previous study in epithelial-like TC-7 cells has shown that MAP7 binds microtubules dynamically with a ~ 4 s recovery half-life ($t_{1/2}$) (*Bulinski et al., 2001*). Similarly, the fluorescent recovery of the N-EGFP fragment occurred with a $t_{1/2}$ of 2.6 s (*Figure 4A,B*), which is equivalent to a dissociation rate constant ($k_{off}$) of $0.27 \pm 0.06$ s$^{-1}$ (*Figure 4B,C*), suggesting that N-EGFP dissociates rapidly after binding to microtubules. Interestingly, however, MAP7-FL-EGFP appears to have >43× tighter microtubule binding, with a longer half-life ($t_{1/2}$ = 112.6 s) of fluorescence recovery and a slower $k_{off}$ ($0.0062 \pm 0.0020$ s$^{-1}$) (*Figure 4A–C*). Also, MAP7-FL remained bound to microtubules with a total recovery of $38 \pm 7\%$, whereas the N fragment showed a large mobile fraction with a total recovery of $74 \pm 3\%$ (*Figure 4D*). The difference in binding affinity appears to be contributed by the P domain, as the ΔC-EGFP fusion also has a longer half-life of recovery ($t_{1/2}$= 59.0 s) and lower $k_{off}$ ($0.012 \pm 0.003$ s$^{-1}$) than the N fragment alone (*Figure 4C,D*, *Figure 4—figure supplement 1A*). Consistent with this, the P-EGFP fusion protein has a half-life of recovery ($t_{1/2}$ of 39.6 s) and a mobile fraction ($92 \pm 9\%$ total recovery), which are similar to those of the ΔC fragment (*Figure 4B,C,D*). The kinetic difference between N and P fragments is further confirmed by the two deletion mutants: ΔP-EGFP and ΔN-EGFP, which exhibited a sixfold difference in recovery half-life ($t_{1/2}$) and $k_{off}$ (*Figure 4—figure supplement 1B,C*), despite having similar amounts of mobile fraction (*Figure 4—figure supplement 1D*). Thus, MAP7 has two unique microtubule binding domains with distinct dissociation kinetics. Taken together, our studies have identified a new microtubule binding site in the P domain that contributes to tight microtubule binding by MAP7 in cells.

## MAP7 enhances kinesin-1 recruitment to microtubules

We next asked how MAP7 binding to microtubules affects kinesin-1 localization in cells. We tested this in COS cells, which express low levels of endogenous kinesin-1 (*Cai et al., 2007*). First, we tested the constitutively active kinesin-1 motor, KIF5C(1-560), which contains the motor head domain and the stalk1 domain (*Norris et al., 2015*). The protein was tagged with a monomeric fluorescent protein mNeonGreen (mNeGr) (*Shaner et al., 2013*) at the C-terminus, and when expressed in live COS cells, the fusion protein (KIF5C(1-560)-mNeGr) appeared predominantly cytosolic, with occasional decoration of microtubules (*Figure 5A*). Interestingly, when MAP7-FL-mCherry was co-expressed, nearly all microtubules were decorated with KIF5C(1-560)-mNeGr (*Figure 5A*). This can be demonstrated by line scans of the KIF5C(1-560)-mNeGr fluorescence intensity across microtubules in the cell (*Figure 5A,B*). In the presence of MAP7-FL, the fluorescent intensity in the

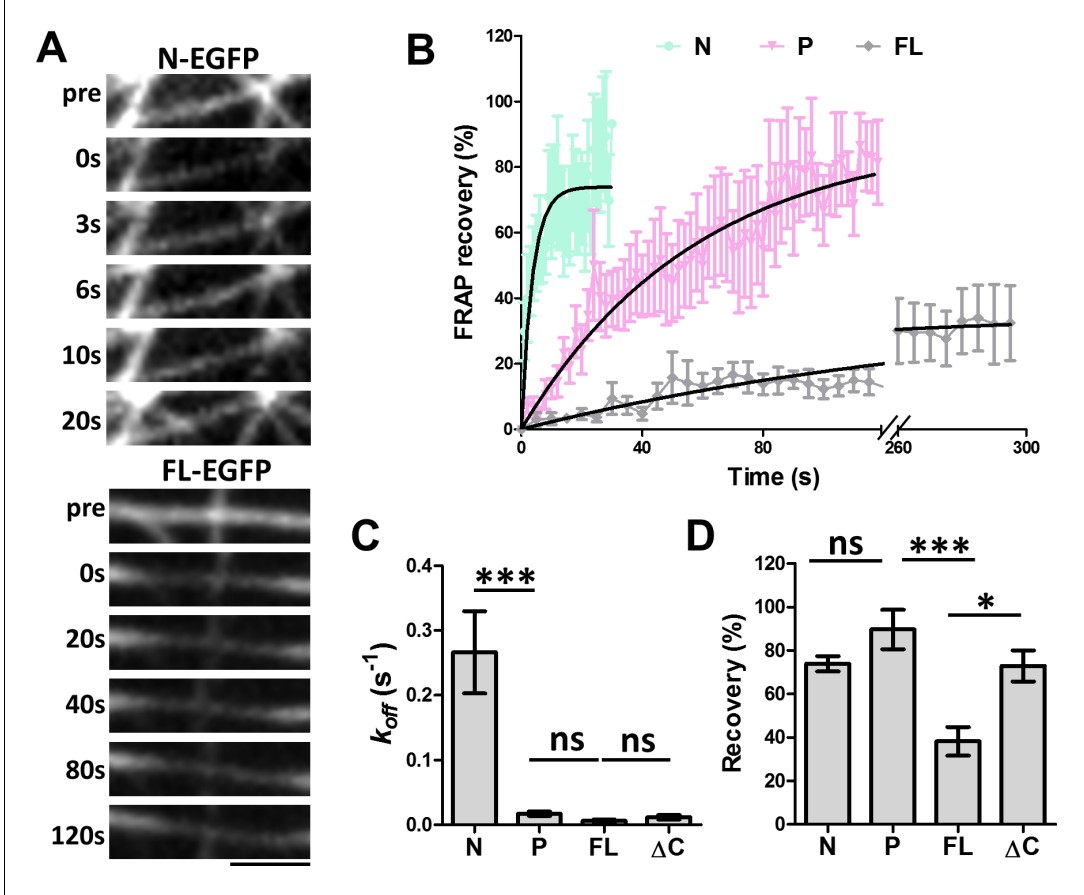

**Figure 4.** FRAP analysis of MAP7 binding to microtubules in COS cells. (**A**) Representative sequential confocal images of N-EGFP or FL-EGFP on single microtubules in live COS cells before (pre) and after photobleaching at various time points. (**B**) Fluorescence recovery plots for N-, P-, and FL-EGFP fusion proteins along single microtubules in COS cells. Solid lines are derived from non-linear curve fitting based on the inverse of an exponential decay model. (**C, D**) Comparison of the dissociation rate constant ($k_{off}$) (**C**) and the total recovery (**D**) derived from curve fitting of the MAP7 N-, P-, FL-, and ΔC-EGFP fluorescence recovery. n = 321 for N (11 cells), 396 for P (9 cells), 420 for FL (7 cells), and 150 for ΔC (5 cells). ANOVA-test (Mean ±SEM) for $k_{off}$ (**C**): N-P, p=0.001; P-FL, p=0.99; FL-ΔC, p=0.99, and for recovery (**D**): N-P, p=0.39; P-FL, p=0.001; FL-ΔC, p=0.018. *p<0.05; ***p<0.001; ns: not significant. Scale bar: 2 μm.

DOI: https://doi.org/10.7554/eLife.36374.010

The following source data and figure supplements are available for figure 4:

**Source data 1.** Data for the fluoresence recovery curve (**Figure 4B**) and the kinetic parameters (**Figure 4C,D**).

DOI: https://doi.org/10.7554/eLife.36374.013

**Figure supplement 1.** FRAP analysis of MAP7 binding to microtubules in COS cells.

DOI: https://doi.org/10.7554/eLife.36374.011

**Figure supplement 1—source data 1.** Data for the fluoresence recovery curve (A,B) and the kinetic parameters (C,D) in **Figure 4—figure supplement 1**.

DOI: https://doi.org/10.7554/eLife.36374.012

peak, corresponding to the microtubule region, is more than two times higher than that in the baseline, which represents the fluorescence in the cytoplasm (**Figure 5C**). Such a difference is much smaller in control cells, which have nearly uniform KIF5C(1-560)-mNeGr fluorescence intensities (**Figure 5B,C**). To demonstrate that kinesin-1 indeed binds to microtubules, we triple transfected COS cells with KIF5C(1-560)-mNeGr, mCherry-tagged tubulin (**Shaner et al., 2004**), and MAP7-FL fused with iRFP670, a far-red fluorescent protein derived from bacterial phytochromes (**Shcherbakova and Verkhusha, 2013**). As shown in **Figure 5D**, both MAP7-FL-iRFP and KIF5C(1-560)-mNeGr are present on the same microtubules. Thus, MAP7 can promote kinesin recruitment to microtubules in cells.

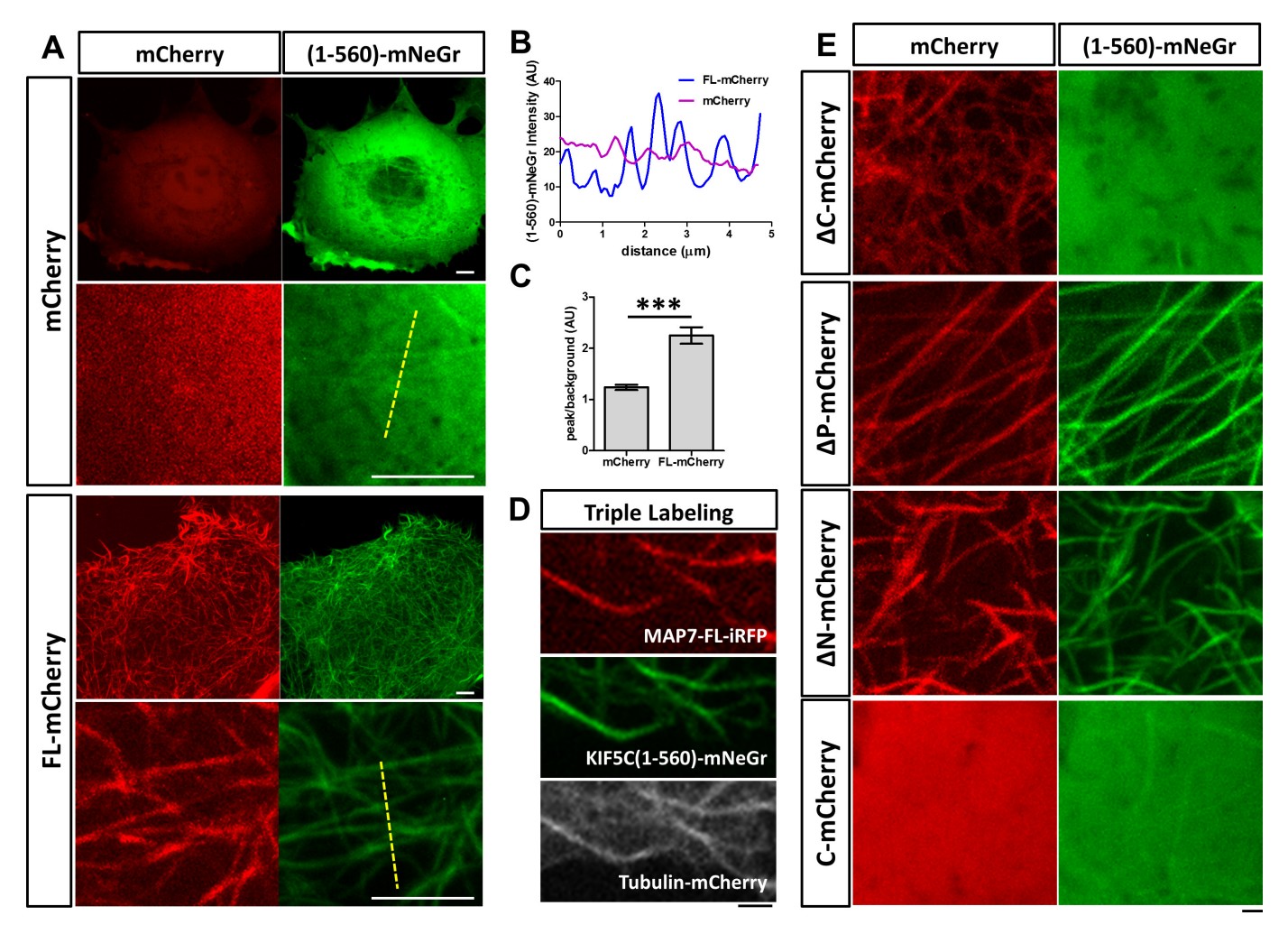

**Figure 5.** MAP7 enhances kinesin-1 binding to microtubules in COS cells. (**A**) Confocal images of live COS cells expressing KIF5C(1-560)-mNeGr (green) with mCherry or MAP7-FL-mCherry (red). High magnification images are shown in the bottom panel of each condition. (**B, C**) Line scan analysis (**B**) of the KIF5C(1-560)-mNeGr fluorescence intensity from the dashed lines in (**A**), and comparison of the peak/background signals from the mNeGr fluorescence in cells expressing mCherry or MAP7-FL-mCherry (**C**). ***p=0.0004 from t-test (Mean ±SEM). n = 5 lines from one cell. (**D**) Confocal images of a region in a live COS cell triple transfected to express MAP7-FL-iRFP (red), KIF5C(1-560)-mNeGr (green), and tubulin-mCherry (white). (**E**) High magnification confocal images of COS cells expressing KIF5C(1-560)-mNeGr (green) with mCherry fusion proteins (red) of the indicated MAP7 fragments.

DOI: https://doi.org/10.7554/eLife.36374.014

The following source data is available for figure 5:

**Source data 1.** Line scan data and quantification of peak/background ratio in *Figure 5B,C*.
DOI: https://doi.org/10.7554/eLife.36374.015

We next investigated the MAP7 domain that is required for recruiting kinesin-1 to microtubules. Early studies have shown that the CC2-containing C domain of MAP7 interacts with the kinesin-1 iso-form KIF5B (*Metzger et al., 2012*). Consistent with this finding, the ΔC fragment lacking the C domain was unable to promote KIF5C(1-560) decoration of microtubules (*Figure 5E*), despite its own tight association with microtubules (*Figures 5E* and *4D*). Furthermore, the microtubule binding sites of MAP7 are required for the interaction between kinesin-1 and microtubules, as the C domain alone did not recruit kinesin-1 to microtubules (*Figure 5E*). Interestingly, only one microtubule bind-ing site is needed, as both ΔP and ΔN can promote kinesin recruitment (*Figure 5E*).

A previous study has suggested that the MAP7-kinesin-1 interaction is mediated by the stalk1 domain of KIF5B (*Metzger et al., 2012*). To confirm this for the kinesin-1 isoform KIF5C, we tested the role of the motor head domain (1-335) and the stalk1 domain (335-560) (*Figure 6A*). A fusion protein containing only the motor domain (KIF5C(1-335)-mNeGr) showed cytosolic expression when co-expressed with MAP7 (*Figure 6B*), suggesting that the stalk1 domain is required for recruitment. However, a fusion protein containing only the stalk1 domain (KIF5C(335-560)-mNeGr) was recruited to microtubules in the presence of MAP7 (*Figure 6B*), suggesting that the stalk1 domain is sufficient for the kinesin-1 interaction with MAP7 and the subsequent kinesin-1 recruitment to microtubules.

To understand the dynamics of the interaction between MAP7 and kinesin-1 on microtubules, we carried out FRAP analysis in COS cells co-expressing MAP7-FL-iRFP with the mNeGr fusions of different kinesin-1 domains. MAP7-FL-iRFP binds to microtubules similarly to the EGFP fusion protein with a low total recovery (<15%) and a long half-life (t = 411 s), indicative of a stable interaction with microtubules (*Figure 6C,D*). In contrast, the two stalk1-containing proteins, KIF5C(1-560)-mNeGr and KIF5C(335-560)-mNeGr, associated dynamically with MAP7-decorated microtubules (*Figure 6D*). Both showed similar dissociation kinetics with 60-70% total recoveries (*Figure 6E*; *Figure 6F*), but the motor head domain (KIF5C(1-560)-mNeGr) has a longer half-life ($t_{1/2}=9$ s) and a slower $k_{off}$ ($0.08 \pm 0.01$ $s^{-1}$) than the stalk1 domain (KIF5C(335-560)-mNeGr) ($t_{1/2} = 5$ s and $k_{off} = 0.14 \pm 0.02$ $s^{-1}$) (*Figure 6E*). These results indicate that while the stalk1 domain alone can mediate kinesin-1 recruitment to microtubules via its binding with MAP7, the motor domain prolongs the kinesin-1 association with microtubules, presumably mediated by its ATP-dependent motor function. Thus, our studies provide the first evidence of dynamic recruitment of kinesin-1 to microtubules by MAP7 in cells.

## MAP7 affects kinesin-1-mediated organelle transport in COS cells

One function of microtubules is to provide tracks for intracellular transport of organelles, such as mitochondria. As MAP7 enhances kinesin-1 recruitment to microtubules, we next asked how this interaction affects organelle transport in cells. To address this question, we employed a recently developed optogenetic approach to transiently recruit kinesin-1 motors to mitochondria (*van Bergeijk et al., 2015*). COS cells were transfected to express two fusion proteins (*Figure 7— figure supplement 1*): a 560-PDZ fusion protein that has the active motor KIF5C(1-560)-mNeGr fused to a PDZ peptide (*Strickland et al., 2012*), and a TOM-LOV fusion that has a mitochondrial targeting sequence (TOM) fused to mCherry and a LOV domain (*van Bergeijk et al., 2015*). Blue light (488 nm) stimulation causes a conformational change in the LOV domain, which results in heterodimerization of the LOV domain and the PDZ peptide, thereby loading the active kinesin motor (560-PDZ) to the mitochondrial surface via TOM-LOV. The resulting movement of KIF5C(1-560)-loaded mitochondria provides a convenient readout for the active kinesin-1 motor-mediated transport. As previously shown (*van Bergeijk et al., 2015*), TOM-LOV mitochondria were largely immobile and localized to the perinuclear region of COS cells, but were transported from the cell center to the cell periphery upon blue light stimulation (*Figure 7—figure supplement 2A*). This movement depends on the expression of the engineered kinesin-1 motor fusion protein, as mitochondria showed little or no movement when 560-PDZ was replaced with the maxGFP control (*Figure 7—figure supplement 2B*).

To assess how MAP7 affects kinesin-mediated transport, 560-PDZ and TOM-LOV were expressed in COS cells along with MAP7-FL-iRFP or iRFP as a control. Upon light activation, the TOM-LOV-labeled mitochondria moved from the central region to the periphery over a 30-min time period, irrespective of MAP7-FL-iRFP or iRFP expression (*Figure 7—figure supplement 2A*), suggesting that MAP7 does not affect overall mitochondrial movement. We therefore analyzed the behavior of individual mitochondria using fast acquisition rates (1–2 frames per second) immediately after blue light stimulation. During the 2-min imaging window, the majority of TOM-LOV-labeled mitochondria remained in the perinuclear region (*Figure 7A*), but some displayed clear directional movements to the cell periphery (*Figure 7B*). We analyzed their movement by projecting the time-lapse movie with each frame color-coded (*Figure 7C*) and using the trajectories to generate kymographs (*Figure 7D*). Each track on the kymograph was then divided into segments of directional movements and segments of pauses (*Figure 7D*).

From the kymograph, we measured the run length and run time of the motile segments and then calculated the average run speed. We found that MAP7 decreased the speed by 29% (0.50 ± 0.06

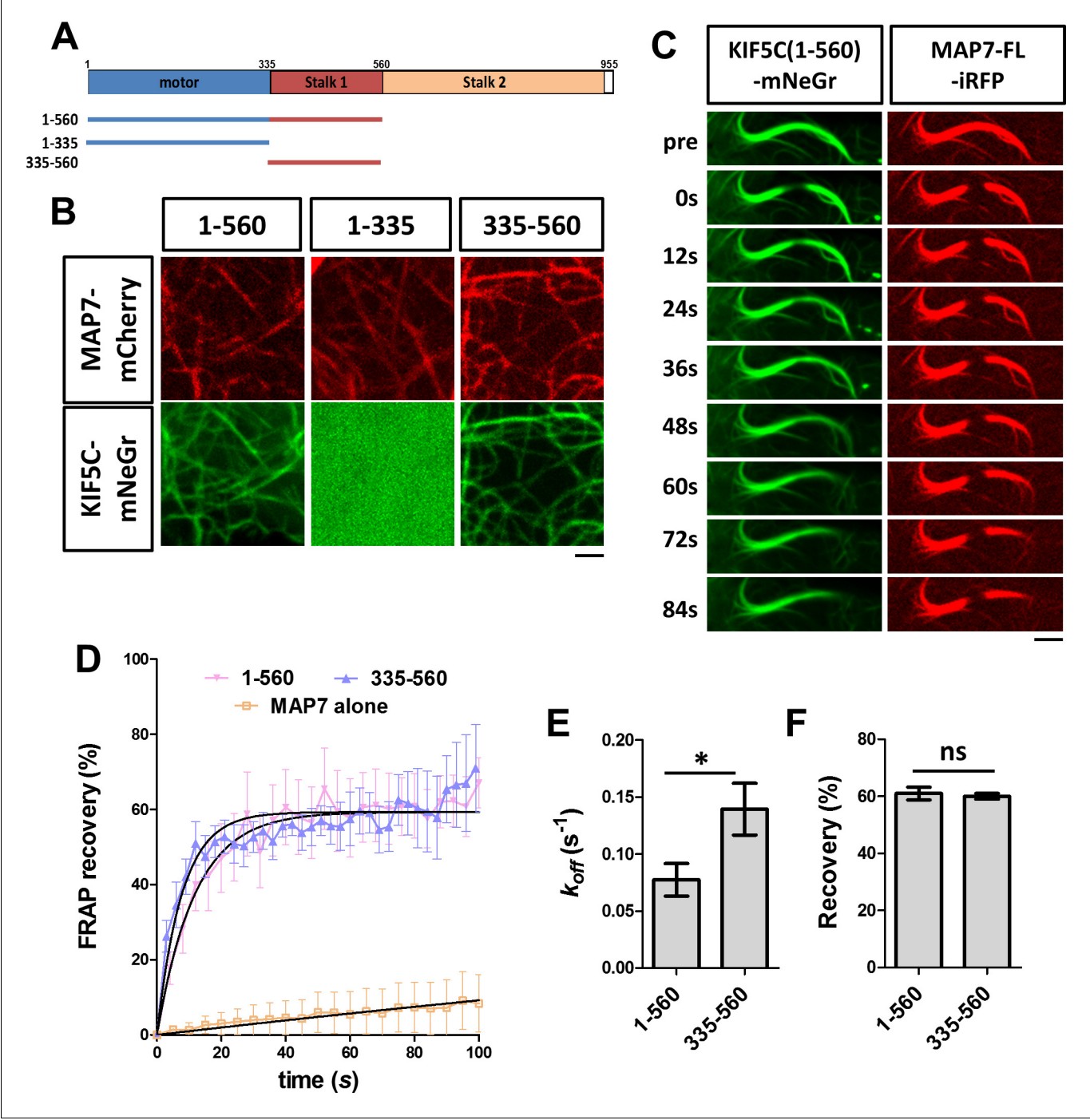

**Figure 6.** Analysis of kinesin-1 domains for its recruitment to MAP7-bound microtubules. (**A**) Domain structure of KIF5C, which includes motor head, stalk1, and stalk2 domains. The structures of various truncated constructs used in this study are illustrated by line drawings. (**B**) Confocal images of live COS cells expressing MAP7-FL-mCherry (red) with mNeGr fusions of the indicated KIF5C domains. (**C**) Sequential confocal images of KIF5C(1-560)-mNeGr (green) and MAP7-FL-iRFP (red) in live COS cells before (pre) and after photobleaching at various time points. (**D**) FRAP recovery plots for the mNeGr fusion proteins of the active motor KIF5C(1-560) and the stalk1 domain KIF5C(335-560), as well as MAP7-FL-iRFP. Solid lines are derived from non-linear curve fitting. (**E, F**) Comparison of the off-rate constant ($k_{off}$) and the total recovery derived from curve fitting for the mNeGr fusion proteins of KIF5C(1-560) and KIF5C(335-560) on MAP7-FL-iRFP-bound microtubules. n = 182 for KIF5C(1-560) and 238 for KIF5C(335-560) from seven cells each condition. T-test (Mean ±SEM) for $k_{off}$ (**E**): p=0.03 and for recovery (**F**): p=0.65. *p<0.05; and ns: not significant. Scale bars: 2 μm.

DOI: https://doi.org/10.7554/eLife.36374.016

The following source data is available for figure 6:

*Figure 6 continued on next page*

*Figure 6 continued*

**Source data 1.** Data for the fluoresence recovery curve (D) and the kinetic parameters (E,F) in *Figure 6*.
DOI: https://doi.org/10.7554/eLife.36374.017

μm/s versus 0.70 ± 0.07 μm/s in controls cells, *Figure 7E*). Next, we analyzed the pause behavior and found that MAP7-FL-iRFP did not significantly alter the pause time (10.4 ± 0.8 s versus 11.9 ± 1.0 s in control cells). Interestingly, however, there was a large increase (36%) in the pause frequency (4.8 ± 0.6 events/min versus 3.1 ± 0.3 events/min in control cells) (*Figure 7F*). The change in pausing behaviors prompted us to examine the moving mitochondria trajectories more carefully. In the color-coded projection containing the space-time information, we found that motile mitochondria in control cells usually resumed relatively straight movement after a pause (hashes in *Figure 7C, D* and *Figure 7—figure supplement 2C*), resulting in linear or slightly curved trajectories. However, motile mitochondria in MAP7-expressing cells often resumed movement after pauses with sharp turns (with angles larger than 90° along the trajectory) (asterisks in *Figure 7C,D*, *Figure 7—figure supplement 2C*). We quantified this behavior and found that MAP7-expressing cells had 1.6-fold more sharp turns (2.44 ± 0.28 events/min) than control expressing cells (1.39 ± 0.13 events/min) (*Figure 7G*). Such sharp directional changes are unlikely to occur along single microtubules, but rather are likely to result from TOM-LOV mitochondria switching tracks onto neighboring or crossing microtubules.

To understand whether these changes depend on the interaction between MAP7 and kinesin-1, we examined MAP7-ΔC-iRFP that does not bind KIF5C(1-560) and found that it had little effect on the speed (0.64 ± 0.06 μm/s, *Figure 7E*). It also showed no statistical difference in the pause frequency (2.5 ± 0.5 events/min, *Figure 7F*) or the sharp turn frequency (0.98 ± 0.19 events/min, *Figure 7G*) when compared with those of the control. These results indicate that the effects of MAP7 on active kinesin-1-mediated mitochondrial movement are dependent on the interaction between the C domain and kinesin-1.

To test whether these observations depend only on kinesin-1 or also on the organelle being transported, we used the same approach to target peroxisomes with a fusion protein (PEX-LOV) containing a peroxisome targeting sequence (PEX) attached to mCherry and the LOV domain (*van Bergeijk et al., 2015*). Like mitochondria (*van Bergeijk et al., 2015*), peroxisomes are relatively immobile in COS cells, as expression of the PEX-LOV alone without 560-PDZ showed no movement (data not shown). However, light activation of COS cells co-expressing 560-PDZ and PEX-LOV led to movement of PEX-LOV-labeled peroxisomes (*Figure 7—figure supplement 3*). We thus analyzed the same parameters described above and found that the effect of MAP7 on pausing is similar to that found with mitochondria. MAP7-FL-iRFP had no significant change in the run speeds (0.26 ± 0.02 μm/s versus 0.31 ± 0.01 μm/s in controls, *Figure 7H*), but increased the pause frequency by 60% (5.7 ± 0.7 events/min versus 3.5 ± 0.5 events/min in controls, *Figure 7I*) as well as the sharp turn frequency by 61% (2.54 ± 0.36 times/min versus 1.56 ± 0.19 times/min in controls, *Figure 7J*). Also, there was no significant change in the pause duration (13.5 ± 2.04 times/min versus 12.9 ± 1.29 times/min in control cells). As a comparison, MAP7-ΔC-iRFP showed no significant change in velocity (0.30 ± 0.01 μm/s), pause frequency (2.53 ± 0.5 events/min), or sharp turn frequency (0.91 ± 0.19 times/min) (*Figure 7H*; *Figure 7I*; *Figure 7J*). Thus, these optogenetic results in COS cells have identified a consistent effect of MAP7 on the pause or track switching behavior of kinesin-1-mediated organelle transport.

## MAP7 affects kinesin-1-mediated organelle transport in DRG axons

To examine how MAP7 influences kinesin-1-mediated organelle transport in neuronal cells, we used the same optogenetic approach to activate 560-PDZ-mediated TOM-LOV mitochondria in the axons of E17 DRG neurons (*Figure 8A*). We used kymographic analysis to compare mitochondrial movement in regions along the axon with high MAP7-FL-iRFP with that in similar regions of control neurons expressing only iRFP (*Figure 8B,C*). When we examined the impact of MAP7 on the optogenetically activated mitochondrial movement in axons co-expressing 560-PDZ, we found a 36% reduction in the average run speed, from 0.55 ± 0.07 μm/s in control cells to 0.35 ± 0.03 μm/s in MAP7-expressing cells (*Figure 8D*). Similarly, there was a decrease in the speed seen in MAP7-Δ

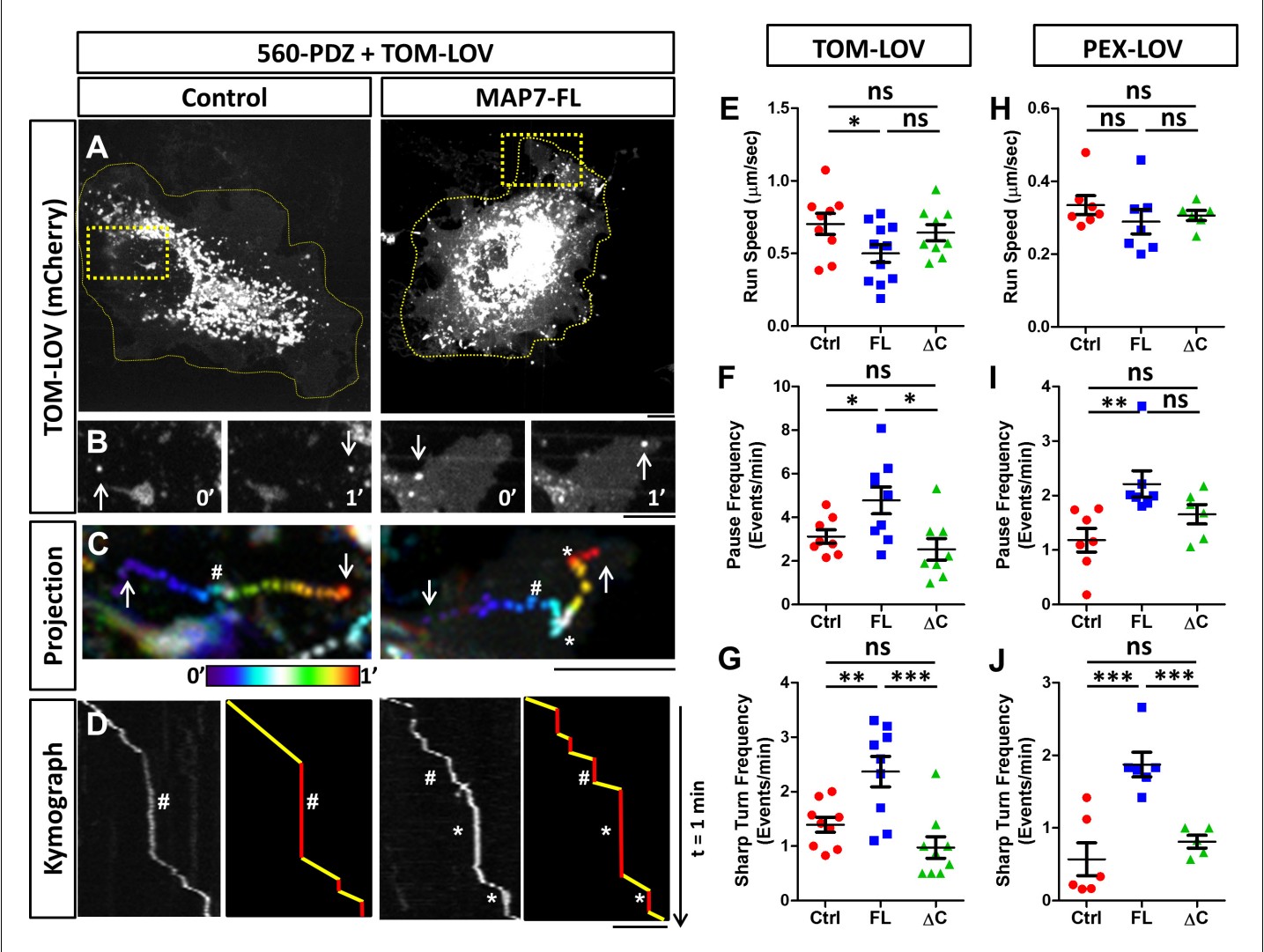

**Figure 7.** MAP7 affects kinesin-1-mediated transport in COS cells revealed by optogenetic activation of organelle movement. (**A, B**) Analysis of TOM-LOV-labeled mitochondrial movement in COS cells expressing TOM-LOV and 560-PDZ together with the control iRFP or MAP7-FL-iRFP. High magnification TOM-LOV images of the boxed areas in (**A**) are shown at time 0' and at time 1' in (**B**). Arrows indicate the mitochondria analyzed below in (**C, D**). (**C**) Sequential images at 1-s intervals are projected with temporal color-coding to show the trajectories of motile mitochondria. Arrows indicate the beginning and the end of each trajectory. The two ends of the color bar represent 0' and 1' respectively. Asterisks (*) indicate pause followed by sharp turns and hashes (#) indicate pauses followed by straightruns. (**D**) Kymographs (left panel) of TOM-LOV-labeled mitochondria are generated based on the trajectories in (**C**) andeach run track is represented by line drawing (right panel) to display regions of movement (yellow) and regions of pausing (red). Asterisks (*) and hashes (#) indicate corresponding pauses in (**C**). (**E–G**) Comparisons of mitochondria run speed (**E**), pause frequency (**F**), and sharp turn frequency (**G**) between cells expressing TOM-LOV, 560-PDZ, and the iRFP control (ctrl), MAP7-FL-iRFP (FL), or MAP7-ΔC-iRFP (ΔC). n = 8 cells (>4 tracks per cell). T-test (Mean ±SEM) for (**E**): Ctrl-FL, p=0.04; Ctrl-ΔC, p=0.52; FL-ΔC, p=0.10; for (**F**): Ctrl-FL, p=0.04; Ctrl-ΔC, p=0.32; FL- ΔC, p=0.67; and for (**G**): Ctrl-FL, p=0.007; Ctrl-ΔC, p=0.10; FL-ΔC, p=0.0009. (**H–J**) Comparisons of peroxisome run speeds (**H**), pause frequency (**I**), and sharp turn frequency (**J**) between cells expressing PEX-LOV, 560-PDZ, and the iRFP control (ctrl), MAP7-FL-iRFP (FL), or MAP7-ΔC-iRFP (ΔC). n = 8 cells (>4 tracks per cell). T-test (Mean ±SEM) for (**H**): Ctrl-FL, p=0.31; Ctrl-ΔC, p=0.37; FL-ΔC, p=0.01; for (**I**): Ctrl-FL, p=0.008; Ctrl-ΔC, p=0.13; FL-ΔC, p=0.10; and for (**J**): Ctrl-FL, p=0.001; Ctrl-ΔC, p=0.38; FL- ΔC, p=0.006. *p<0.05, **p<0.005, ***p<0.001 and ns: not significant. Scale bars: 5 μm.

DOI: https://doi.org/10.7554/eLife.36374.018

The following source data and figure supplements are available for figure 7:

**Source data 1.** Data for the comparasion of several movemen parameters in *Figure 7E–J*.
DOI: https://doi.org/10.7554/eLife.36374.022
**Figure supplement 1.** Construction of protein domains used in the optogenetic experiments.
DOI: https://doi.org/10.7554/eLife.36374.019
**Figure supplement 2.** Optogenetic activation of kinesin-1-mediated mitochondrial movement in COS cells.

*Figure 7 continued*

DOI: https://doi.org/10.7554/eLife.36374.020

**Figure supplement 3.** Optogenetic activation of kinesin-1-mediated peroxisome movement in COS cells.
DOI: https://doi.org/10.7554/eLife.36374.021

C-iRFP (0.39 ± 0.03 μm/s). However, as in COS cells, the change in speed was specific to mitochondria. When we expressed 560-PDZ-mediated PEX-LOV peroxisome movement, we found no significant changes in the run speeds by MAP7-FL and MAP7-ΔC (*Figure 8E*, *Figure 8—figure supplement 1A*, control: 0.46 ± 0.05 μm/s; MAP7-FL: 0.40 ± 0.03 μm/s; MAP7-ΔC: 0.37 ± 0.06 μm/s). Furthermore, we used MitoTracker to test the effect of MAP7 on the behavior of endogenous kinesin-1-mediated mitochondrial movement (*Figure 8—figure supplement 2B,C*) and found no significant effect of overexpressed MAP7-FL or MAP7-ΔC in E17 DRG neurons (*Figure 8F*; Control: 0.30 ± 0.02 μm/s; MAP7-FL: 0.35 ± 0.03 μm/s; MAP7-ΔC: 0.37 ± 0.02 μm/s) or in E14 DRG neurons (*Figure 8G*, control: 0.40 ± 0.03 μm/s; MAP7-FL: 0.32 ± 0.03 μm/s; MAP7-ΔC: 0.33 ± 0.04 μm/s). Thus, MAP7 affects the speed of mitochondrial movement mediated only by the active kinesin-1 motor but not by the endogenous kinesin-1 in axons.

In contrast to transport speed, we found a more consistent effect of MAP7 on pausing or track switching in axons. Unlike COS cells, TOM-LOV-labeled mitochondria in axons rarely pause, but instead, they undergo changes of speeds during each run segment (*Figure 8B,C*, arrows in green/yellow lines). We thus calculated the frequency of speed switching and compared it with the pausing behavior seen in COS cells. Upon light activation, MAP7 expression increased the speed change frequency for 560-PDZ-driven mitochondria by 2.1-fold, from 1.28 ± 0.19 events/min in controls to 2.3 ± 0.33 events/min in MAP7-FL-iRFP axons (*Figure 8D*). This effect depends on the MAP7-kinesin-1 interaction, as there was no impact with MAP7-ΔC-iRFP on the speed change (*Figure 8D*; 0.77 ± 0.21 events/min). The same effect was also found for 560-PDZ-driven peroxisomes, as co-expression of PEX-LOV and MAP7-FL-iRFP significantly increased the number of speed change events when compared with both the iRFP control and MAP7-ΔC-iRFP (*Figure 8E*, *Figure 8—figure supplement 1A*; control: 0.8 ± 0.17 events/min; MAP7-FL: 2.12 ± 0.24 events/min; MAP7-ΔC: 1.56 ± 0.38 events/min). These results suggest that the effect of MAP7 on speed switching is not limited by the type of organelle once they are loaded with the active kinesin-1 motor.

To determine whether the effect applies to endogenous kinesin-1, we analyzed MitoTracker-labeled mitochondria movement. Again, we found an increase in speed change events elicited by MAP7-FL-EGFP but not by MAP7-ΔC-EGFP in E17 DRG axons (*Figure 8F*, *Figure 8—figure supplement 1B*; control: 0.98 ± 0.13 events/min; MAP7-FL: 2.04 ± 0.22 events/min; MAP7-ΔC: 0.77 ± 0.20 events/min), suggesting that endogenous kinesin-1 behaved similarly to the active KIF560 motor. This effect is consistent with results from the control experiments done with TOM-LOV-labeled mitochondria without the active motor KIF-PDZ. Here, MAP7 increased movement switching by twofold, from 1.26 ± 0.33 events/min in control axons to 3.12 ± 0.36 events/min, whereas MAP7-ΔC showed no change (0.64 ± 0.23 events/min; *Figure 8—figure supplement 2*). Finally, we also examined MitoTracker-labeled mitochondria in E14 DRG axons, which do not have endogenous MAP7 expression. Again, overexpression of MAP7-FL-EGFP but not MAP7-ΔC-EGFP increased the speed switching frequency (*Figure 8G*; *Figure 8—figure supplement 1C*, control: 1.13 ± 0.13 events/min; MAP7-FL: 1.54 ± 0.16 events/min; MAP7-ΔC: 1.11 ± 0.23 events/min), confirming the role of MAP7-FL. Taken together, analysis in DRG axons suggests that MAP7 influences speed switching in kinesin-1-mediated organelle movement. This function depends on the kinesin-1 interaction and likely reflects track switching seen in COS cells.

## Discussion

The ability of MAP7 to interact with both microtubules and kinesin suggests unique regulatory mechanisms for MAP7 function in diverse neuronal processes. Our detailed domain analysis of MAP7 in DRG neurons has revealed distinct roles of MAP7 domains in regulating axon growth and branching. Analysis of MAP7 interactions with microtubules and kinesin-1 has also mapped out the role of each domain in these interactions and suggested a potential function of MAP7 in regulating kinesin-1-

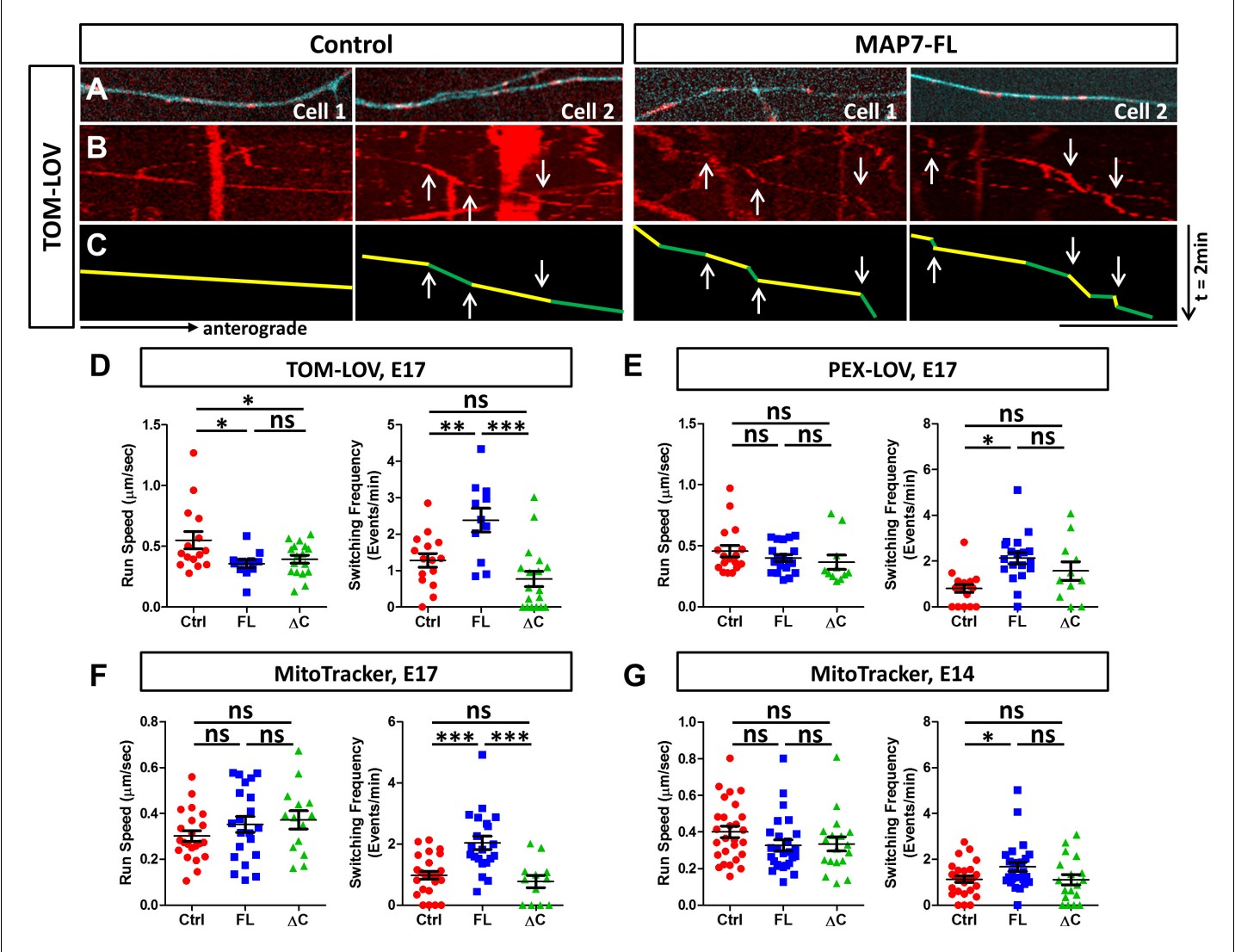

**Figure 8.** MAP7 induces changes in kinesin-mediated organelle movement in DRG axons. (**A**) Representative images of TOM-LOV-labeled mitochondria (red) from DRG axons in cells expressing 560-PDZ and TOM-LOV with either the iRFP control or MAP7-FL-iRFP (cyan). (**B, C**) Kymographs (**B**) for the images shown in (**A**), and line drawings of representative runs with yellow and green segments of varying speeds (**C**). Changes in speed along a run are indicated by arrows. (**D, E**) Comparison of mitochondrion (**D**) or peroxisome (**E**) run speed and frequency of speed switching between E17 DRG axons expressing TOM-LOV (**D**) or PEX-LOV (**E**) along with 560-PDZ and the iRFP control (Ctrl), MAP7-FL-iRFP (FL), or MAP7-ΔC-iRFP (ΔC). n ≥ 11 cells per condition and n > 5 runs for each cell. T-test (Mean ±SEM) for (**D**): run speed: Ctrl-FL, p=0.04; Ctrl-ΔC, p=0.04; FL-ΔC, p=0.47; switching frequency: Ctrl-FL, p=0.005; Ctrl-ΔC, p=0.08; FL-ΔC, p=0.0002; and for (**E**): run speed: Ctrl-FL, p=0.30; Ctrl-ΔC, p=0.23; FL-ΔC, p=0.55; switching frequency: Ctrl-FL, p=0.0001; Ctrl-ΔC, p=0.06; FL-ΔC, p=0.21. (**F, G**) Comparison of run speed and speed switching frequency of MitoTracker-labelled mitochondria between E17 (**F**) or E14 (**G**) DRG neurons expressing the EGFP control (Ctrl), MAP7-FL-EGFP(FL), or MAP7-ΔC-EGFP(ΔC). n = 16 cells (>2 tracks per cell). T-test (Mean ±SEM) for (**F**): run speed: Ctrl-FL, p=0.23; Ctrl-ΔC, p=0.11; FL-ΔC, p=0.71; switching frequency: Ctrl-FL, p=0.001; Ctrl-ΔC, p=0.39; FL-ΔC, p=0.005; and for (**G**): run speed: Ctrl-FL, p=0.09; Ctrl-ΔC, p=0.19; FL-ΔC, p=0.88; switching frequency: Ctrl-FL, p=0.03; Ctrl-ΔC, p=0.93; FL-ΔC, p=0.076. *p<0.05, ***p<0.001 and ns: not significant. Scale bar: 10 µm.

DOI: https://doi.org/10.7554/eLife.36374.023

The following source data and figure supplements are available for figure 8:

**Source data 1.** Data for the comparison of run speed and switching frequency in *Figure 8D–G*.
DOI: https://doi.org/10.7554/eLife.36374.026
**Figure supplement 1.** Behaviors of optogenetically activated peroxisome movement or endogenous MitoTracker-labeled mitochondrial movement in DRG axons.
DOI: https://doi.org/10.7554/eLife.36374.024
**Figure supplement 2.** Comparisons of run speed and frequency of speed changes of TOM-LOV mitochondria in DRG axons.
*Figure 8 continued on next page*

*Figure 8 continued*

DOI: https://doi.org/10.7554/eLife.36374.025

**Figure supplement 2—source data 1.** Data for the comparison of run spped and switching frequency in *Figure 8—figure supplement 2*.

DOI: https://doi.org/10.7554/eLife.36374.027

mediated organelle transport. Based on these data, we propose a mechanism involving these interactions in regulating organelle transport during axon morphogenesis.

## Different roles of MAP7 binding to microtubules and kinesin-1 in axon growth, branch formation, and branch growth

Our previous study (*Tymanskyj et al., 2017*) showed that FL-MAP7 is localized to branch sites to promote branch maturation. The current study maps out the domains of MAP7 responsible for this localization as well as for axon growth and branching (*Figure 1*). Both N and P domains are responsible for branch formation, whereas the C-terminal kinesin-interacting domain is important for promoting axon and branch growth. As both N and P domains bind microtubules and are both required for localizing MAP7 proteins to branch sites, MAP7 could potentially regulate microtubule organization seen at the branch sites (*Ketschek et al., 2015*).

Interestingly, the newly identified microtubule binding site in the P domain (see discussion below) is sufficient to promote axon growth in cooperation with the kinesin-interacting C domain, as demonstrated by the difference in axon growth between ΔP-EGFP and ΔN-EGFP (*Figure 1*). The key difference between these two truncated proteins is that ΔN has a higher microtubule binding affinity than ΔP. As both ΔN and ΔP can recruit kinesin-1 to microtubules (*Figure 6*), this result suggests that a tight microtubule binding site, along with the potential phosphorylation regulation in the P domain, is required for the kinesin-1 function in axon growth.

Although the kinesin interaction mediated by the C domain is dispensable for branch formation, it is required for branch growth, as demonstrated by shorter branches in neurons expressing the ΔC fragment that lacks the C domain when compared with MAP7-FL-expressing neurons (*Figure 1*). The fact that MAP7-FL-expressing neurons have shorter main axons than control cells (*Figure 1K*), suggests that branch growth mediated by the C domain can compete for main axon growth. In the absence of branch formation, the C domain is required for the main axon growth, as shown by the ability of the ΔN fragment but not the P domain alone to promote axon elongation. These results not only support the role of the kinesin-interacting domain in axon and branch growth but also reveal how MAP7 localization affects its function. Therefore, our domain analysis has revealed distinct roles of MAP7 interactions with microtubules and kinesin-1 in different steps of axon morphogenesis.

## The P domain of MAP7 harbors a second microtubule binding site critical to tight MAP7-microtubule interaction

Detailed domain analysis of MAP7 binding to microtubules in COS cells has revealed several features of this interaction. First, previous studies based on deletion analysis have suggested that microtubule binding is mediated by CC1 in the N domain, which contains 13 positively charged residues (*Faire et al., 1999*; *Masson and Kreis, 1993*). Using FRAP analysis, we were able to demonstrate that this domain binds to microtubules transiently with fast dissociation kinetics. Our analysis also suggests that MAP7 is composed of two microtubule binding domains, with the second microtubule binding site residing in the P domain, as deletion of the N domain does not prevent microtubule binding but the P domain itself is sufficient to bind microtubules (*Figure 3A*). The P domain contains 26 positively charged residues, suggesting that it might bind microtubules directly. Indeed, the *in vitro* co-sedimentation analysis with purified proteins supports this property (*Figure 3B*).

In addition, the FRAP analysis has shown that MAP7 binds microtubules in COS cells with slow dissociation kinetics, which is manifested by a low total recovery and a long recovery half-life (*Figure 4*). Thus, MAP7 has a more stable microtubule binding property than tau ($t_{1/2}$ =~4 s) (*Sayas et al., 2015*). The slow MAP7 dissociation kinetics are similar to those obtained recently using a single molecule assay *in vitro* (*Monroy et al., 2018*) but differ from an earlier study in TC-7 cells, which shows a shorter half-life (*Bulinski et al., 2001*). The difference may be a result of the different

cell lines or MAP7 fusions used. It is interesting to note that our analysis of the N domain did show faster microtubule dissociation kinetics, similar to the previous study (*Bulinski et al., 2001*).

Furthermore, the P domain binds microtubules more tightly than the N domain (*Figure 4*) and is likely to contribute to the kinetic property of the MAP7-FL protein. In addition, deletion of either the N or the P domain alters the binding kinetics, suggesting that both domains cooperate to regulate microtubule structures that are critical to branch formation. As MAP7 is phosphorylated during mitosis, likely at many potential sites within the P domain (*Masson and Kreis, 1995*; *Ramkumar et al., 2018*; *Mchedlishvili et al., 2018*), this newly found microtubule binding site could be a target for regulation in different cellular processes.

## MAP7 enhances kinesin-1 recruitment to microtubules

We have demonstrated for the first time that co-expression of MAP7 and kinesin-1 leads to recruitment of kinesin-1 to microtubules in COS cells. This is consistent with the previous demonstration of their functional and physical interactions in several systems (*Barlan et al., 2013*; *Gallaud et al., 2014*; *Metzger et al., 2012*; *Sung et al., 2008*). In addition, we not only confirmed that this interaction is mediated by the C domain but also demonstrated that it requires at least one microtubule binding site in either the N or P domain (*Figure 5*).

Domain analysis of kinesin-1 suggests that the stalk1 domain of KIF5C is both necessary and sufficient for this interaction, consistent with the previous co-immunoprecipitation data (*Metzger et al., 2012*). Furthermore, FRAP analysis reveals that the interaction between the stalk1 domain and MAP7 is transient, with a half-life of 5 s, which could support a controlling mechanism of kinesin-mediated motility discussed below. Interestingly, the kinesin-1 motor head that binds microtubules in an ATP-dependent manner can influence the kinetics (*Figure 6*), suggesting that other regions of the kinesin-1 molecule (*Figure 6A*) may affect its interaction with MAP7 on microtubules. In fact, the FL kinesin-1 molecule is normally folded with the motor domain inhibited by a C-terminal inhibitory peptide (*Cai et al., 2007*; *Hackney and Stock, 2000*; *Verhey and Hammond, 2009*), and the tail region of kinesin-1 contains a sequence that can bind microtubules directly (*Hackney and Stock, 2000*; *Seeger and Rice, 2010*). Since the FL kinesin-1 molecule can be also recruited to microtubules by MAP7 *in vitro* (*Sung et al., 2008*), microtubule-bound MAP7 may therefore unfold kinesin-1 and stimulate motility, as suggested by the recent study of *Drosophila* MAP7 in S2 cells (*Barlan et al., 2013*).

## MAP7 regulates kinesin-1-mediated organelle transport by modulating pause/speed switching

What is the functional role of the kinesin-1 recruitment to microtubules by MAP7? Our analysis provides clues for a direct impact of MAP7 on kinesin-1-mediated transport. One effect of MAP7 expression seemed to decrease the run speed of 560-PDZ-bound mitochondria in both COS and neurons. However, when comparing the MAP7 effect on mitochondria driven by endogenous kinesin-1 or 560-PDZ-bound peroxisomes, we did not see a similar change in speed. The different effects may reflect the number of motors present on the organelle surface under each condition and/or additional roles of the kinesin-1 stalk2 and tail domains in the endogenous FL kinesin-1 function. In neurons, this speed difference is independent of the C domain of MAP7, suggesting that binding of MAP7 to microtubules, possibly because of a crowding effect, may be responsible for reducing the speed of kinesin-1-mediated transport along microtubules.

More interestingly, one consistent effect of MAP7 in COS cells and DRG axons is the increase in pause/speed switching during organelle movement. This was found for optogenetically activated mitochondrial or peroxisome movement in both COS cells and axons as well as for the endogenous mitochondrial movement in axons (*Figures 7,8*). In COS cells, we noticed that this switching correlated with mitochondria changing microtubule tracks, as demonstrated by prominent sharp turns in MAP7 expressing cells (*Figure 7*). This behavior is similar to early *in vitro* observations for other kinesin molecules (*Ross et al., 2008*; *Schroeder et al., 2012*) and suggests that MAP7 bound to neighboring microtubules may enhance kinesin-bound cargos switching to those microtubules. In fact, *in vitro* kinesin-1 motors often disengage from microtubules completely when encountering multiple microtubules, but on occasions can engage to a different microtubule (*Ross et al., 2008*). It is likely that MAP7-bound microtubules, with an increased capability to recruit kinesin-1, can increase the

likelihood of kinesin-1 binding to neighboring microtubules and thus redirect the kinesin-bound cargos. As microtubules are highly bundled in axons (*Kapitein and Hoogenraad, 2015*; *Mikhaylova et al., 2015*), the speed switching elicited by MAP7 in axons observed here may be reminiscent to the track change in COS cells and suggests a potential MAP7-mediated mechanism for ensuring efficient kinesin-1 engagement and transport within the axon.

Our data show that unlike other MAPs (*Encalada and Goldstein, 2014*), MAP7 influences kinesin-1 motility by enhancing motor recruitment to microtubules and increasing the frequency of pausing or speed switching. Although our analysis centers mainly on mitochondria, which normally uses kinesin-1 for movement (*Glater et al., 2006*; *Hurd and Saxton, 1996*; *Lin and Sheng, 2015*), the same effects may apply to other kinesin-1-mediated transport as suggested by our study of peroxisomes. Thus, we propose a mechanism by which microtubule-bound MAP7 promotes axon or branch growth by recruiting kinesin-1 to microtubules and modulating organelle transport. When MAP7 is concentrated at branch junctions via interactions of its N and P domains with microtubules, it will slow down or pause moving organelles at that junction and facilitate track switching, thereby diverting organelles into branches. This effect leads to a reduction of organelle transport into the main axon and shifts the balance from main axon growth to branch growth. Consistent with this possibility, mitochondria are often associated with branch sites (*Armijo-Weingart and Gallo, 2017*; *Courchet et al., 2013*; *Spillane et al., 2013*). In the absence of branch formation, however, microtubule-bound MAP7 recruits kinesin-1 and drives organelle movement in the main axon such as seen with ΔN, thereby increasing axonal length. Thus, the presence of MAP7 at branch sites and its ability to recruit kinesin-1 to microtubules and subsequently to switch organelles to different microtubules are essential to MAP7 function in regulating branch formation. Such functions of MAP7 mediated by its dual interactions with microtubules and kinesin-1 may provide a mechanism for other morphogenetic processes (*Barlan et al., 2013*; *Gallaud et al., 2014*; *Metzger et al., 2012*; *Sung et al., 2008*) and represent a new mechanism to regulate microtubule-based transport by motor and non-motor MAPs in neurons (*Encalada and Goldstein, 2014*; *Fu and Holzbaur, 2014*; *Maday et al., 2014*; *Nirschl et al., 2017*).

## Materials and methods

**Key resources table**

| Reagent type | Designation | Source or reference | Identifiers | Additional information |
|---|---|---|---|---|
| Strain, strain background (*Rattus norvegicus*) | Sprague Dawley (rat) | Charles River | | |
| Cell line (*Cercopithecus aethiops*) | COS-7 cell (monkey) | Lab stock | RRID:CVCL_0224 | |
| Recombinant DNA reagent | pEGFP-N3 | Clontech | GeneBank #U57609 | |
| Recombinant DNA reagent | MAP7-FL-EGFP | (*Tymanskyj et al., 2017*) | | |
| Recombinant DNA reagent | MAP7-ΔN-EGFP | This paper | | |
| Recombinant DNA reagent | MAP7-ΔC-EGFP | (*Tymanskyj et al., 2017*) | | |
| Recombinant DNA reagent | MAP7-ΔP-EGFP | This paper | | |
| Recombinant DNA reagent | MAP7-N-EGFP | This paper | | |
| Recombinant DNA reagent | MAP7-P-EGFP | This paper | | |

*Continued on next page*

*Continued*

| Reagent type | Designation | Source or reference | Identifiers | Additional information |
|---|---|---|---|---|
| Recombinant DNA reagent | MAP7-C-EGFP | This paper | | |
| Recombinant DNA reagent | GST-MAP7-N | This paper | | |
| Recombinant DNA reagent | MAP7-P-mCherry-His | This paper | | |
| Recombinant DNA reagent | MAP7-C-mCherry-His | This paper | | |
| Recombinant DNA reagent | MAP7-FL-mCherry-His | This paper | | |
| Recombinant DNA reagent | MAP7-ΔC -mCherry-His | This paper | | |
| Recombinant DNA reagent | MAP7-ΔP-mCherry-His | This paper | | |
| Recombinant DNA reagent | MAP7-ΔN -mCherry-His | This paper | | |
| Recombinant DNA reagent | iRFP-N3 | This paper | | |
| Recombinant DNA reagent | MAP7-FL-iRFP | This paper | | |
| Recombinant DNA reagent | MAP7-ΔC-iRFP | This paper | | |
| Recombinant DNA reagent | KIF5C(1-560)-mNeGr | (*Norris et al., 2015*) | | |
| Recombinant DNA reagent | KIF5C(1-335)-mNeGr | This paper | | |
| Recombinant DNA reagent | KIF5C(335-560)-mNeGr | This paper | | |
| Recombinant DNA reagent | TOM-mCherry-LOVpep (TOM-LOV) | (*van Bergeijk et al., 2015*) | | |
| Recombinant DNA reagent | PEX-mCherry-LOVpep (PEX-LOV) | (*van Bergeijk et al., 2015*) | | |
| Recombinant DNA reagent | KIF5C(1-560)-PDZ-mNeGr (560-PDZ) | This paper | | |
| Recombinant DNA reagent | Tubulin-mCherry | (*Shaner et al., 2004*) | | |
| Antibody | α-tubulin (rabbit) | Abcam | # ab18251 RRID:AB_2210057 | 1:5000 |
| Antibody | neurofilament (RMO-270) (mouse) | ThermoFisher Scientific | # 13–0700 RRID:AB_2532998 | 1:1000 |
| Antibody | mCherry (rabbit) | Rockland | # 600-401-379 RRID:AB_2209751 | 1:2500 |
| Antibody | FLAG (rabbit) | Sigma | F7425 RRID:AB_439687 | 1:2500 |
| Antibody | C-terminal MAP7 (rabbit) | GeneTex | # GTX120907 RRID:AB_11170884 | 1:1000 |
| Antibody | GFP (chick) | Aves lab | #GFP-1020 RRID:AB_10000240 | 1:2000 |
| Antibody | Goat anti-Rabbit IgG, Alexa 680 | ThermoFisher Scientific | #A-21109 RRID: AB_2535758 | 1:1000 |
| Antibody | Alexa 488 Donkey anti-chicken IgY (IgG)(H + L) | Jackson ImmunoResearch | #703-545-155 RRID:AB_2340375 | 1:1000 |

*Continued on next page*

Continued

| Reagent type | Designation | Source or reference | Identifiers | Additional information |
|---|---|---|---|---|
| Antibody | Alexa 568 goat anti-rabbit IgG (H + L) | Invitrogen | #A11036 RRID:AB_10563566 | 1:1000 |
| Antibody | Cy3-AffiniPure Donkey Anti-Mouse IgG (H + L) | Jackson ImmunoResearch | #715-165-150 RRID:AB_2340813 | 1:1000 |
| Antibody | Peroxidase-AffiniPure Goat-Anti Mouse IgG | Jackson ImmunoResearch | #115-035-003 RRID:AB_10015289 | 1:2500 |
| Antibody | Peroxidase-AffiniPure Donkey Anti-Chicken IgY (IgG) (H + L) | Jackson ImmunoResearch | 703-035-155, RRID:AB_10015283 | 1:2000 |
| Recombinant proteins | GST-MAP7-N-FLAG | This paper | | |
| Recombinant proteins | MAP7-P-mCherry-His | This paper | | |
| Recombinant proteins | MAP7-C-mCherry-His | This paper | | |
| Commercial assay or kit | P3 primary cell transfection reagent | Lonza | V4XP-3012 | |
| Chemicals | MitoTracker Red CMXRos | ThermoFisher Scientific | M7512 | |
| Chemicals | Clarity Western ECL | BioRad | 1705060 | |
| Software | ImageJ/Fiji | NIH | RRID:SCR_003070 | |
| Software | GraphPad Prism | GraphPad Software, | RRID:SCR_002798 | |

## DNA constructs, cell lines, and animals

MAP7 constructs based on the mouse sequence (BC052637) were created by PCR and confirmed by Sanger sequencing. All fusion proteins were created by subcloning the PCR products into the EcoRI and KpnI sites in frame with the C- or N-terminal tag of the following vectors: 1) EGFP-N3; 2) iRFP-N3; 3) pCAGGS-mCherry-His; and 4) pGEX-6P1-FLAG. The iRFP-N3 vector was generated by replacing EGFP of the EGFP-N3 backbone at the BamHI and NotI sites with a PCR product of iRFP670 from pMito-iRFP670, a gift from Vladislav Verkhusha (Addgene plasmid # 45462) (*Shcherbakova and Verkhusha, 2013*). pCAGGS-mCherry-His was generated by subcloning a PCR product containing KpnI-mCherry and His tag into the pCAGGS-ES vector at the EcoRI and SacI sites. For GST-tagged expression in bacteria, the pGEX-6P1 vector (GE Life sciences, 27-1542-01) was modified by subcloning an oligo containing the KpnI site and the FLAG sequence behind the GST tag using the EcoRI and SalI sites. Tubulin-mCherry (*Shaner et al., 2004*) was a gift from Roger Tsien (UCSD).

The KIF5C(1-560)-mNeGr construct has been described previously (*Norris et al., 2015*). mNeGr fusion proteins of other kinesin-1 domain constructs ((1-335) and (335-560)) were created first by PCR from KIF5C-mCit (*Norris et al., 2015*). After sequence confirmation, the PCR products were subcloned into the EcoRI and KpnI sites by swapping with KIF5C(1-560) in the KIF5C(1-560)-mNeGr backbone. These mNeGr fusion constructs were later subcloned into a pCAGGS-ES vector using the EcoRI and NotI sites.

The TOM-mCherry-LOVpep (TOM-LOV) and PEX-mCherry-LOVpep (PEX-LOV) constructs (gift from Lukas Kapitein) used in the optogenetic experiments have been described previously (*van Bergeijk et al., 2015*). The KIF5C(1-560)-PDZ-mNeGr (560-PDZ) construct was created by subcloning the PDZ peptide behind mNeGr at the BsrGI and Not1 sites in the pCAGGS-KIF5C(1-560)-mNeGr construct. The PDZ peptide was cut from the KIF-PDZ plasmid (gift from Lukas Kapitein) (*van Bergeijk et al., 2015*).

COS-7 cells (originally from the lab stock of the Tessier-Lavigne lab) were grown in DMEM with 10% fetal bovine serum (Invitrogen). They were authenticated based on morphology, and DNA staining revealed no mycoplasma contamination.

Timed pregnant Sprague Dawley rats were obtained from Charles River and used in accordance with the Guidelines for the Care and Use of Laboratory Animals of the National Institutes of Health

and the approved IACUC protocol (#01560) of the Thomas Jefferson University. Vaginal plug dates were designated as E0.

## DRG neuron culture and axon branch analysis

Primary rat DRG neuronal cultures were performed as described previously (*Zhao et al., 2009* )). Briefly, DRGs were dissected out from E14 or E17 rat embryos, washed once in HBSS, and incubated at 37°C with 0.25% trypsin for 10–15 min. Trypsin-treated DRGs were resuspended in L15 medium plus 10% horse serum and then mechanically triturated with a fire-polished glass pipette. Dissociated rat DRG neurons (~7.5 $\times$ 10$^5$ cells) were transfected with various MAP7 constructs by nucleofection (Lonza) using reagent P3 and the CU-133 program. Neurons were then plated at ~30,000 cells on 18 mm glass coverslips coated with 10 µg/mL poly-D-lysine and 10 µg/mL laminin and cultured in F12 medium (with the N3 supplement, 40 mM glucose, and 25 ng/mL NGF). Cells were fixed in 4% PFA/ PBS after 24 hr for immunocytochemical analysis.

EGFP fusion proteins of MAP7 fragments expressed in DRG neurons were visualized directly by the fluorescence of the EGFP tag. Neuronal morphology was visualized by antibody staining of neurofilament. For antibody labeling, PFA-fixed neurons were blocked in PBS plus 5% goat serum and 0.1% Triton (PGT) for 1 hr and then incubated with primary antibodies diluted in PGT for at least 1 hr at room temperature or overnight at 4°C. After washing, they were incubated with Cy3, Alexa 488, or Alexa560-labeled secondary antibodies (Jackson ImmunoResearch or Invitrogen) diluted in PGT for 1 hr at room temperature. Digital images were taken on Zeiss Axiovert 200 microscope with a 20$\times$ or 40$\times$ objective by a Yokogawa spinning disk confocal system using 488, 561, or 642 nm laser excitation and an sCMOS camera (Zyla, Andor). Images are shown as 2D projections and inverted in gray-scale.

Based on the digital confocal images, the following parameters were analyzed in ImageJ: main axon length, number of branches per cell, and branch length. The main axon was defined as the longest axon from the cell body to the growth cone. Branches longer than 10 µm were traced and counted. They were divided into two groups based on the length of the axon. Branches emerging from the distal 10% of the axon were defined as terminal branches and those emerging from the other 90% of the axon were termed interstitial branches and considered as collateral branches in the analysis. For branch length comparison, branches longer than 10 um were measured and included in the analysis. Each condition was repeated more than three times with >20 neurons analyzed.

## Live cell imaging and FRAP analysis of MAP7 and kinesin in COS cells

COS cells were transfected using a TransMax system (gift from Yun Yao) with various constructs after plating on glass bottom dishes (MatTek). After overnight culture, dishes were mounted on a heated humidified chamber (OkoLab) equilibrated to 37°C with 5% $CO_2$ on an inverted microscope (Zeiss Axiovert 200). Fluorescent images were acquired from live cells using a 100$\times$ apochromatic objective (NA = 1.4) and an EMCCD camera (Cascade 512, Photometrics) or an sCMOS camera on a spinning disk system (described above).

For FRAP analysis of MAP7, we selected COS cells expressing low levels of C-terminal tagged EGFP fusion proteins and regions containing single microtubules. For FRAP analysis of kinesin, we selected regions of COS cells that contain microtubule bundles bound with MAP7-FL-iRFP and KIF5C(1-560)-mNeGr. Photobleaching was achieved by 5–10$\times$ pulses of 435 nm pulse laser using an Andor MicroPoint system. EGFP, mNeGr, and iRFP images were acquired by a 100$\times$ objective and an EMCCD camera described above for 1–5 mins at 1–5 s intervals. Cells were cotransfected with a tubulin-mCherry vector to monitor any damage to microtubules from bleaching.

To analyze fluorescent changes in the bleached region, the fluorescent signal was measured from a box placed in three regions: bleached, unbleached, and background. The signal in the background region was first subtracted from that in the bleached and unbleached regions of the same frame. The background subtracted signal in the bleached region was then normalized to the signal in the unbleached region at each time point. The percentage of fluorescence recovery was then determined by calculating the ratio of fluorescence change after bleaching to the fluorescence difference before and after photobleaching using the recovery equation $(F_t - F_0)/(F_{pre}-F_0)$, where $F_t$ is the fluorescence measured at each time point, $F_0$ is the fluorescence signal immediately after photobleaching at $t = 0$ s and $F_{pre}$ is the fluorescence intensity before photobleaching. The recovery (excluding

the number from the out-of-focus frames because of microtubule movement) was plotted and then fit by least square nonlinear regression in Prism (GraphPad) based on the inverse of an exponential decay model with the initial recovery set to 0 at $t = 0$ and the maximum recovery set at 100. The kinetic parameters ($t_{1/2}$ and $k_{off}$) and the total recovery were obtained from the fit, assuming a diffusion-uncoupled binding reaction.

## Recombinant protein purification and co-sedimentation assay

To obtain recombinant P-mCherry or C-mCherry proteins, 120 mL cultures of HEK 293F cells transfected with mCherry-His fusion DNA (in the pCAGGS-mCherry-His vector) were spun down at 5000 rpm, 4°C, for 10 min. Pellets were re-suspended in 2 mL modified RIPA buffer (mRIPA, 10 mM Tris-HCL, pH 8, 140 mM NaCl, 1% Triton-X-100, 0.1% sodium deoxycholate, 0.1% SDS), plus EDTA-free Protease Inhibitor cocktail (Roche) and 1 mM PMSF on ice. Cell suspensions were lysed in a dounce homogenizer and spun for 30 min, 14 k rpm, at 4°C. The supernatant fraction was filtered through a 0.25 μm syringe filter and then incubated with 0.4 mL of 50% nickel bead slurry (BioRad) and allowed to mix end-over-end at 4°C for 2.5 hr. Lysate-bead mixture was added to a 10 mL chromatography column, and the flow through was allowed to rebind 3×. Beads were then washed with 100 mL of wash buffer (50 mM Tris pH 8, 300 mM NaCl, 1 mM PMSF, 0.1 mM EDTA, 5 mM ß-mercaptoethanol (ßME), 5 mM imidazole) to remove contaminants and prevent non-specific binding. Bound MAP7 fusion proteins were eluted off beads by incubating the beads with 0.2 mL elution buffer (50 mM Tris pH 8, 175 mM NaCl, 1 mM PMSF, 1× Protease inhibitor, 0.1 mM EDTA, 5 mM ßME, 500 mM imidazole) for 5 min before collection. Eluted proteins were dialyzed overnight in 800 mL BRB80 (80 mM PIPES, 1 mM EGTA, 1 mM $MgCl_2$ and 5 mM ßME) at 4°C and 10% glycerol was added. Finally, purified proteins were aliquoted and snap-frozen in liquid nitrogen. Protein concentrations were determined by the Bradford assay using BSA as a standard.

To obtain recombinant GST-tagged MAP7-N proteins, 1 L cultures of BL-21 *E. coli* transformed with the GST-MAP7-N expression plasmid (in pGEX-6P1-FLAG) were incubated in a 37°C degree shaker and induced with 0.1 mM IPTG when OD at 595 nm reached 0.6. Cultures were then allowed to shake overnight at 18°C before being pelleted at 4000 RPM, 15 min, at 4°C. Bacterial pellets were re-suspended in lysis buffer (20 mM Tris pH = 8, 250 mM NaCl, 1% Tween-20, 5 mM ßME, 20% NDSB-201, 5 mM EDTA, 5% glycerol) and then sonicated four times for 20–30 s. Sonicated lysates were pelleted down at 15,000 rpm, 30 min, at 4°C. The supernatants were collected and filtered through a 0.2-μm syringe filter. Glutathione beads were washed and equilibrated in lysis buffer. They were mixed with the filtered supernatants and allowed to incubate for 3 hr at 4°C on a plate shaker. Bead-protein mixes were then added to 10 mL chromatography columns. The flow through was collected and then re-bound to the beads for four times. Protein-bound beads were then washed with 1 L of wash buffer (25 mM Tris pH = 8, 250 mM NaCl, 1 mM DTT, 1 mM EDTA, 0.1% Tween-20, and 0.2 mM PMSF) and 150 mL of wash buffer without Tween-20. Finally, proteins were incubated for 30 min with elution buffer (wash buffer plus 20 mM reduced glutathione, minus Tween) and allowed to shake for 30 min before eluent collection. Purified proteins were dialyzed into BRB80 before being snap frozen in liquid nitrogen.

For the co-sedimentation assay, freshly thawed bovine tubulin was added to BRB80 supplemented with fresh DTT (1 mM) and GTP (1 mM) (BRB80D/G) in a centrifuge tube to a final concentration of 2 mg/mL. The tubulin mixture was spun at 60,000 RPM at 4°C for 15 min in a TLA-100.1 rotor to remove tubulin aggregates. The supernatant was collected and incubated for 5 min in a 37°C water bath before the addition of 0.2, 2, and 20 μM of taxol (Sigma) in DMSO sequentially with 5-min incubations between each addition of taxol. Polymerized MT seeds were then spun through an equal volume of cushion consisting of 60% glycerol in BRB80D/G/T (BRB80D/G + 20 μM Taxol) at 60,000 RPM, 15 min, and 34°C to remove un-polymerized tubulin. Freshly thawed recombinant MAP7-fragment proteins were supplemented with 1 mM DTT, 1 mM GTP, 1× protease inhibitor cocktail (Roche), and 20 μM taxol and then spun at 60,000 RPM, 4°C, 15 min in a TLA100.1 to remove aggregates. Supernatants were collected and then adjusted in BRB80D/G/T to 2–20 μM. For microtubule binding, the MAP7 proteins were used to re-suspend microtubule pellets prepared above for microtubule binding. All samples were allowed to incubate at room temperature for 10 min, and then pelleted down at 35,000 RPM, 15 min, at 22°C. To analyze bound proteins, the first half volume of supernatant was collected and diluted in 2× sample buffer while the remaining

supernatant was removed carefully without touching the pellet. Pellets were then incubated with equal volumes of sample buffer diluted with BRB80 plus 1 mM DTT on ice for 30 min. The pellet was then mechanically disrupted by scraping with tweezers before being re-suspended through repeated pipetting. For no-microtubule controls, MAP7 proteins were incubated alone and then spun the same way as above. For pre-binding control, MAP7 proteins were directly diluted in sample buffer. For protein analysis, small aliquots of each condition (pre, supernatant, and pellet) with equal dilution (1:10 in 1× sample buffer for western, and undiluted for Coomassie staining) were fractionated by 10% SDS-PAGE.

## Western blot analysis

Western blots were probed using rabbit anti-FLAG antibody (Sigma, 1:2500), rabbit anti-RFP antibody (Rockland, 1:2500), or chick anti-GFP antibody (Aves Lab, 1:2000) in 4% milk in PBS-T (PBS and 0.1% Tween-20) for 1 hr. After washing three times (5 min each) in PBS-T, blots were incubated in HRP conjugated anti-rabbit IgG antibody (1:2500) or anti-chick IgY (1:2000) for 1 hr, washed for 5 × 5 min in PBS-T, and then incubated with Clarity ECL reagent (BioRad) for 5 min before being exposed for 1 min to HyBlot ES X-ray film (Denville Scientific) or ChemiDoc Imaging System (BioRad).

## Analysis of kinesin-mediated motility in live COS cells and DRG axons

COS cells or DRG neurons cultured on glass bottom dishes (MatTek) were triple transfected with 560-PDZ, TOM-LOV or PEX-LOV, and iRFP-N3, MAP7-FL-iRFP, or MAP7-ΔC-iRFP. In control experiments, maxGFP (Lonza) was used instead of 560-PDZ. After overnight culture in the dark, dishes were mounted on a heated humidified chamber on an inverted microscope and imaged at 100× as described above. To activate heterodimerization, cells were exposed to blue light via 488 nm laser for 5 s and then immediately imaged. For long-term imaging, images were acquired for all three channels (488, 561, 647 nm) every minute for 30 min. For short-term imaging, images were acquired every 1–2 s for 2 min for mitochondria/peroxisome only (mCherry) or every 0.5–1 s for 2 min for both mitochondria and MAP7(mCherry and iRFP). For single channel mitochondrion imaging, MAP7-FL-iRFP and 560-LOV images were also acquired at the first and last time points. To avoid the microtubule bundles sometimes formed with high MAP7 expression, we chose cells expressing low level MAP7-FL-iRFP for the subsequent analysis on single microtubules.

For experiments examining endogenous mitochondria behavior, MitoTracker Red (5 nM) was added to the neuronal culture (transfected with EGFP, MAP7-FL-EGFP, or MAP7-ΔC-EGFP) for 10 min, subsequently washed twice with warm media and left to recover for 30 min before imaging. Images were acquired every 2 s for 2 min.

To analyze mitochondrial/peroxisome movement behavior, time-lapse images of live cells were first projected and pseudo-colored using the Temporal Color-Code plugin to reveal motile trajectories. Segmented lines were then drawn over the trajectories to generate kymographs in Fiji. Tracks on the kymograph are then analyzed by drawing with the segmented line tool. Each track can be divided into multiple segments. A segment is defined as a portion of a track with a constant speed. It is bound by changes in speed or pauses. Run time (t) and run length (l) were obtained from each segment, and used to calculate the run speed (s) using the formula $s = l/t$. Pauses were defined as any movement slower than 0.1 µm/s. Pause or speed switches were defined as any change in speed along a single track. Mitochondria/peroxisomes which showed no movement in the time window were excluded from run analysis. In COS cells, the directional change was mainly identified in the trajectories within the cell and defined as abrupt changes in directionality with angles more than 90°..

## Statistical analysis

All measurements are presented as mean ±SEM. Statistical analysis was performed in Prism 5.0 software (GraphPad). Normally distributed data (tested by the Kolmogorov-Smirnov) were compared by unpaired two-tailed independent $t$-tests for two samples or one-way ANOVA with Tukey's post hoc analysis for more than three samples. p-Values smaller than 0.05 are considered significant and represented by asterisks described in figure legends.

## Acknowledgments

We are indebted to Lucas Kapitein for sharing the plasmid DNAs used for the optogenetic experiments. We thank Eric Kostuk (Iacovitti lab) and members of the Dalva lab for sharing rat tissues, Gino Cingolani for guidance on protein purification, and Jeff Benovic for sharing the ultracentrifuge. We also thank Bridget Curran and Matthew Dalva for helpful discussion and Tim Mosca and Martin Hruska for comments on the manuscript. This work was supported by grants from NIH to KJV (GM070862) and LM (NS062047), a CURE grant from the Pennsylvania Department of Health to LM (SAP#4100048728), and a Jefferson Synaptic Biology Center grant from Thomas Jefferson University to LM.

## Additional information

### Funding

| Funder | Grant reference number | Author |
|---|---|---|
| National Institute of General Medical Sciences | GM070862 | Kristen J Verhey |
| National Institute of Neurological Disorders and Stroke | NS062047 | Le Ma |
| Pennsylvania Department of Health | SAP # 4100068728 | Le Ma |
| Thomas Jefferson University | Jefferson Synaptic Biology Center | Le Ma |

The funders had no role in study design, data collection and interpretation, or the decision to submit the work for publication.

### Author contributions

Stephen R Tymanskyj, Data curation, Formal analysis, Investigation, Visualization, Methodology, Writing—original draft, Writing—review and editing; Benjamin H Yang, Data curation, Validation, Investigation, Visualization, Writing—original draft; Kristen J Verhey, Resources, Writing—review and editing; Le Ma, Conceptualization, Supervision, Funding acquisition, Investigation, Visualization, Writing—original draft, Project administration, Writing—review and editing

### Author ORCIDs

Stephen R Tymanskyj http://orcid.org/0000-0002-6837-0644
Benjamin H Yang http://orcid.org/0000-0002-9845-0514
Kristen J Verhey http://orcid.org/0000-0001-9329-4981
Le Ma http://orcid.org/0000-0003-2769-9416

### Ethics

Animal experimentation: The study was performed in strict accordance with the Guidelines for the Care and Use of Laboratory Animals of the National Institutes of Health and the approved IACUC protocol (#01560) of the Thomas Jefferson University.

### Decision letter and Author response

Decision letter https://doi.org/10.7554/eLife.36374.030
Author response https://doi.org/10.7554/eLife.36374.031

## Additional files

### Supplementary files

• Transparent reporting form
DOI: https://doi.org/10.7554/eLife.36374.028

## Data availability

All quantitative data for statistical analysis shown in figures are provided as source data in corresponding Excel sheets.

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
