## [Decision Letter]

Thank you for submitting your article "MAP7 Regulates Axon Morphogenesis by Recruiting Kinesin-1 to Microtubules and Modulating Organelle Transport" for consideration by *eLife*. Your article has been reviewed by Marianne Bronner as the Senior Editor, a Reviewing Editor, and two reviewers. The reviewers have opted to remain anonymous.

The reviewers have discussed the reviews with one another and the Reviewing Editor has drafted this decision to help you prepare a revised submission.

Summary:

Both reviewers agree that the model proposed in this manuscript is interesting. Both reviewers found that many aspects of this manuscript are well done. The observations of MAP7 localization, MAP7's recruitment of kinesin-1 are striking. They also raised some technical issues, particularly relating to the mitochondria experiments in Figure 7. We agree that these experiments are important and needs to be strengthened. See below for detailed comments. I would like to invite you to address the reviewers’ comments and submit a revised manuscript.

Essential revisions:

1) We would love to see an effect of reducing endogenous MAP7 on mitochondrial transport in neurons, or some other proxy of kinesin-1 function (such as a different kinesin-1 cargo), as this would strengthen the paper considerably.

2) Figure 1: The explanation of the domain analysis data in Figure 1 is very clear. The only place that was a bit confusing was the description of the cells "we took advantage of the E14 rat DRG neurons that lack endogenous MAP7 expression." These are wild-type DRGs that because of their age happen not to express MAP7? I am guessing this is the correct interpretation, but I initially thought that these were from homozygous mutant mice, so it might be better to make this explanation clearer.

3) Figure 2: Is there any way to quantitate the localization data? As it is, all the reader has are the few example images. How representative are they? How many cells had these patterns? Can the pattern itself be quantitated?

4) In the part of the paper that uses optogenetics to control mitochondrial motility, it would be helpful to add some controls using MAP7 that is unable to bind kinesin-1, but that can still interact with microtubules.

5) For the mitochondrial motility experiments in DRG neurons, the authors use E17 DRG neurons, when MAP7 expression level is at its peak based on their previous paper. More MAP7 is added by overexpression, but wouldn't the results be cleaner if E14 DRGs that do not express MAP7 were used as in as in Figure 1 and Figure 2? The E17 neurons would be a perfect place to look for effects of MAP7 loss.

6) In Figure 7A, where is the motility quantified? MAP7 is only at branch point but mitochondria are everywhere so one might expect different mitochondrial behaviors in different places.

[Editors' note: further revisions were requested prior to acceptance, as described below.]

Thank you for resubmitting your work entitled "MAP7 Regulates Axon Morphogenesis by Recruiting Kinesin-1 to Microtubules and Modulating Organelle Transport" for further consideration at *eLife*. Your revised article has been favorably evaluated by Marianne Bronner (Senior Editor), a Reviewing Editor, and two reviewers.

The manuscript has been improved but there are some remaining issues that need to be addressed before acceptance, as outlined below:

1) The text should be read through carefully for small errors. For example, in the first sentence of the Introduction the subject and verb do not agree.

2) The writing can still be improved to convey a linear series of events. The images in Figure 1B-I have numerous "dots and blobs", and readers would surely not know where to look. Maybe the authors should trace the neurons and only show the tracings (not the images). Figures should be organized based on themes, and every single panel or graph should not be given a different letter.

---

## [Author Response]

Summary:Both reviewers agree that the model proposed in this manuscript is interesting. Both reviewers found that many aspects of this manuscript are well done. The observations of MAP7 localization, MAP7's recruitment of kinesin-1 are striking. They also raised some technical issues, particularly relating to the mitochondria experiments in Figure 7. We agree that these experiments are important and needs to be strengthened. See below for detailed comments. I would like to invite you to address the reviewers’ comments and submit a revised manuscript.

We thank the reviewers for their positive comments of our study as well as their constructive critiques on the technical issues related to the mitochondria experiments. We have carried out additional experiments as suggested to address these issues. The results, explained in more detail below, strengthen the conclusion of the paper.

Essential revisions:1) We would love to see an effect of reducing endogenous MAP7 on mitochondrial transport in neurons, or some other proxy of kinesin-1 function (such as a different kinesin-1 cargo), as this would strengthen the paper considerably.

We agree that this is an important question to address. We thus performed the same set of optogenetic experiments using peroxisomes in both COS cells (Figure 6) and neurons (Figure 7). These studies with PEX-LOV labeled peroxisomes nicely compliment the findings seen with TOM-LOV labeled mitochondria under different experimental conditions. In fact, the results are consistent with regards to the pause or speed switching for both organelles. However, the studies also showed a difference in terms of run parameters, such as run speed (see below). Thus, these new experiments with peroxisomes provide another way to identify and validate common functions of MAP7.

In addition, we have carried out studies using neurons isolated from a gene-trap mouse line that lacks MAP7 expression. The data obtained supports the findings shown in this study.

2) Figure 1: The explanation of the domain analysis data in Figure 1 is very clear. The only place that was a bit confusing was the description of the cells "we took advantage of the E14 rat DRG neurons that lack endogenous MAP7 expression." These are wild-type DRGs that because of their age happen not to express MAP7? I am guessing this is the correct interpretation, but I initially thought that these were from homozygous mutant mice, so it might be better to make this explanation clearer.

We have re-written this sentence to make the difference of E14 rat DRG neurons clearer (subsection “MAP7 domains have distinct roles in axon growth and branching”).

3) Figure 2: Is there any way to quantitate the localization data? As it is, all the reader has are the few example images. How representative are they? How many cells had these patterns? Can the pattern itself be quantitated?

To demonstrate the increased localization of MAP7 to branch sites, we calculated the percentage of branches that have endogenous or overexpressed MAP7 enriched at branch junctions. The data presented in Figure 2D (Final Figure 2B) are discussed in subsection “MAP7 domains have distinct roles in axon growth and branching”.

To further demonstrate the specific MAP7 localization at branch sites, we included line scans of EGFP and tubulin signals in axons expressing MAP7-FL and MAP7-ΔN. The data shown in Figure 2Q,R (Final Figure 2D) are discussed in subsection “MAP7 domains have distinct roles in axon growth and branching”.

4) In the part of the paper that uses optogenetics to control mitochondrial motility, it would be helpful to add some controls using MAP7 that is unable to bind kinesin-1, but that can still interact with microtubules.

We agree with the reviewers that this is a very critical experiment to test specificity. We thus analyzed MAP7-ΔC, which displays similar microtubule binding properties as MAP7-FL but lacks the ability to interact with kinesin. Comparison between MAP7-FL and MAP7-ΔC would test whether the phenotypes we obtained were due to the unique microtubule binding properties of MAP7 or the ability of MAP7 to interact with kinesin-1. We have analyzed this construct in both COS cells and neurons. We have also included the construct in the new peroxisome analysis in COS cells and neurons. Finally, we added this analysis in the study of two different embryonic ages using MitoTracker. The results from these analyses (presented in Figure 6 (Final Figure 7) and Figure 7 (Final Figure 8)) support the role of the C domain of MAP7 in mediating MAP7 regulation of organelle transport, especially pause/speed switching. Thus, these control experiments with MAP7-ΔC provide better insights into the function of MAP7.

5) For the mitochondrial motility experiments in DRG neurons, the authors use E17 DRG neurons, when MAP7 expression level is at its peak based on their previous paper. More MAP7 is added by overexpression, but wouldn't the results be cleaner if E14 DRGs that do not express MAP7 were used as in as in Figure 1 and Figure 2? The E17 neurons would be a perfect place to look for effects of MAP7 loss.

We appreciate the suggestion to examine MAP7 function in E14 neurons that do not express endogenous MAP7. We thus analyzed the effect of MAP7 on MitoTracker-labeled mitochondria between E14 and E17 neurons (Figure 7Q-T (Fignal Figure 8F,G)) and found consistent effects between the two ages. In addition, the analyses in DRG neurons were done in axonal areas with MAP7 expression. Thus, the effects found with E17 neurons do reflect MAP7 function in transport regulation.

*6) In Figure 7A, where is the motility quantified? MAP7 is only at branch point but mitochondria are everywhere so one might expect different mitochondrial behaviors in different places.*

When overexpressed, MAP7-FL is often present in the mid-region of an axon, in addition to its normal branch site localization. Our analysis was done in these axonal regions with detectable MAP7 signals. We have included a clearer description in the text (subsection “MAP7 affects kinesin mediated mitochondrial transport in DRG axons”). We have also included sample images of the transfected axon regions used to generate the kymographs (Figure 7A,B,G,H (Final Figure 7A)).

[Editors' note: further revisions were requested prior to acceptance, as described below.]

The manuscript has been improved but there are some remaining issues that need to be addressed before acceptance, as outlined below:1) The text should be read through carefully for small errors. For example, in the first sentence of the Introduction the subject and verb do not agree.2) The writing can still be improved to convey a linear series of events. The images in Figure 1 B-I have numerous "dots and blobs", and readers would surely not know where to look. Maybe the authors should trace the neurons and only show the tracings (not the images). Figures should be organized based on themes, and every single panel or graph should not be given a different letter.

Following your suggestion, we have edited the manuscript in several areas:

1) We have traced the neuronal morphology in Figure 1 and include them in Figure 1—figure supplement 1.

2) We have re-numbered the figures so images from the same theme are grouped together.

3) We have split previous Figure 3 to new Figure 3 and Figure 4, and then renumbered the rest of the figures.

4) We have updated the text and Source data to reflect the changes in figure numbering.

5) We have updated the Materials and methods section to ensure the consistency of all the reagents used.

6) We have carefully read the text to remove any typos and grammatical errors.